# Unexpected worker mating and colony-founding in a superorganism

Mingsheng Zhuang [1,2,7], Thomas J. Colgan [3,7], Yulong Guo[1], Zhengyi Zhang[1], Fugang Liu[1], Zhongyan Xia[1], Xueyan Dai [2], Zhihao Zhang[1], Yuanjian Li[1], Liuhao Wang[4], Jin Xu[1], Yueqin Guo[1], Yingping Qu[1], Jun Yao[1], Huipeng Yang[1], Fan Yang[1], Xiaoying Li[1], Jun Guo[5], Mark J. F. Brown [6,8] ✉ & Jilian Li [1,8] ✉

The emergence of caste-differentiated colonies, which have been defined as 'superorganisms', in ants, bees, and wasps represents a major transition in evolution. Lifetime mating commitment by queens, pre-imaginal caste determination and lifetime unmatedness of workers are key features of these animal societies. Workers in superorganismal species like honey bees and many ants have consequently lost, or retain only vestigial spermathecal structures. However, bumble bee workers retain complete spermathecae despite 25-40 million years since their origin of superorganismality, which remains an evolutionary mystery. Here, we show (i) that bumble bee workers retain queen-like reproductive traits, being able to mate and produce colonies, underlain by queen-like gene expression, (ii) the social conditions required for worker mating, and (iii) that these abilities may be selected for by early queen-loss in these annual species. These results challenge the idea of lifetime worker unmatedness in superorganisms, and provide an exciting new tool for the conservation of endangered bumble bee species.

More than a century ago, William Morton Wheeler proposed that social insect colonies can be defined as superorganisms when they have morphologically differentiated reproductive and nursing castes that are analogous to the metazoan germ-line and soma[1–3]. More recently, the presence of a lifetime mating commitment by queens (i.e., mating occurs only after emergence, followed by sperm storage for life), pre-imaginal caste differentiation (mapping on to the Wheeler reproductive and nursing castes), and lifetime worker unmatedness have been proposed as both pre-requisites for, and the definition of, superorganisms[2,3]. Once entire offspring cohorts are defined by lifetime unmatedness, which results from their preimaginal differentiation into workers, the lineage has passed its point of no return from an

organismal level of organization to that of a superorganism[2,3], marking a major transition in evolution[4]. In haplodiploid superorganisms, males are haploid and females, both queens and workers, are diploid[2,3]. Worker reproduction in honey bees, some stingless bees, and the majority of ant species is regulated not only by behavioral and physiological constraints, but also by morphological and functional constraints in the reproductive organs[5,6]. Specifically, the degeneration or loss of reproductive structures, such as ovaries and the spermatheca, is widespread in ants, honey bees, and stingless bees[7]. This not only removes the cost of developing and maintaining these organs in unmated individuals, but such morphological specializations between queens and workers would also promote efficient division of labor with

[1]State Key Laboratory of Resource Insects, Institute of Apicultural Research, Chinese Academy of Agricultural Science, Beijing, China. [2]Shanghai Suosheng Biotechnology Co., Ltd, Shanghai 201700, China. [3]Institute of Organismic and Molecular Evolution, Johannes Gutenberg University Mainz, 55128 Mainz, Germany. [4]College of Resources and Environmental Sciences, Henan Institute of Science and Technology, Xinxiang, Henan 453003, China. [5]Faculty of Life Science and Technology, Kunming University of Science and Technology, Kunming, Yunnan 650500, China. [6]Centre for Ecology, Evolution and Behaviour, Department of Biological Sciences, School of Life Sciences and the Environment, Royal Holloway University of London, Egham, UK. [7]These authors contributed equally: Mingsheng Zhuang, Thomas J. Colgan. [8]These authors jointly supervised this work: Mark JF Brown, Jilian Li. ✉e-mail: Mark.Brown@rhul.ac.uk; bumblebeeljl@hotmail.com

reduced conflicts within a colony[8]. As a consequence, once workers in these groups have evolved into obligate helpers, they cannot produce female offspring[9,10]. However, in the superorganismal bumble bees, vespine wasps and certain ant subfamilies, unmated females (workers) retain an intact spermatheca[11–13]. Their occurrence suggests that they have been actively maintained by selection, but the fitness rewards, if any, remain elusive. The retention of apparently intact and energetically expensive reproductive systems in bumble bee workers since superorganismality evolved in this taxon 25–40 million years ago[14] remains a mystery.

Here, using a series of integrated experiments, we show that bumble bee workers of multiple species retain a functional spermatheca, enabling them to produce diploid offspring. Artificially inseminated workers express the behavior of queens, founding and producing colonies that run through the entire lifecycle. This is underpinned by queen-like gene expression. We demonstrate through a series of behavioral experiments that workers are attractive to males and have the ability to mate if they have experienced social isolation. De-queening of colonies triggers this behavior, suggesting that mated workers might take-over colonies in the wild after queen loss. Not only do these results challenge the concept of lifetime unmatedness of developmentally-determined workers in superorganisms, but they provide a novel tool for the conservation of endangered bumble bee species.

## Results

### Assessment of bumble bee worker spermatheca functionality

To test whether the spermatheca of bumble bee workers is functional, that is, can receive, store, and release sperm to fertilize eggs, we artificially inseminated[15] 30 workers of *Bombus lantschouensis* and compared them to 30 non-inseminated control workers. Twenty-three inseminated workers produced female offspring (Fig. 1a). In comparison, 26 non-inseminated workers also laid eggs but produced only male offspring (Fig. 1a), as would be expected for normal reproductively active bumble bee workers[9,10]. To determine if such patterns were species-specific, we repeated this experiment with two additional bumble bee species, *B. ignitus* (which is in the same subgenus as *B. lantschouensis*, *Bombus sensu stricto*, with a common ancestor ~8MYA[14]) and *B. montivagus* (which is in the subgenus *Megabombus*, sharing a common ancestor with the subgenus *Bombus s.s.* ~ 25MYA[14]), which both responded similarly to the artificial insemination treatment by producing female offspring (Fig. 1b, c). As these species diverged from a common ancestor ~25 MYA[14], the most parsimonious explanation of our findings suggest that such abilities and associated mechanisms may be conserved across all social bumble bees.

### Can bumble bee workers act as functional reproductive queens?

To understand if bumble bee workers can act as functional queens (i.e., rear a colony through the complete lifecycle), we used the buff-tailed bumble bee, *B. terrestris* (also a member of the subgenus *Bombus s.s.*, which shares a common ancestor with *B. ignitus* ~ 6MYA[14]), as a model system to compare colony production by artificially inseminated workers and queens. While 94% of all workers (113/120) and 97% of all queens (29/30) successfully laid eggs ~8–9 days after treatment (Fig. 1d, e), only 83% of workers (25/30) and 90% of queens (27/30) that went through artificial insemination and received sperm produced female offspring (Fig. 1e). Microsatellite analysis confirmed that these female offspring of each worker-produced colony were diploid and fathered by the male whose sperm had been used for artificial insemination (Supplementary Data 1–6). Colonies produced by workers and queens went through the same stages: 1st brood, 2nd brood cells constructed on 1st brood pupae, continuous brood production, and an eventual switch to haploid egg-laying, including onset of the competition phase (Fig. 1f). However, worker-produced colonies passed through these colony stages more quickly after the first eclosion of

workers (worker vs. queen (mean ± SD): 1st brood: 34.8 ± 3.34 days vs 36.8 ± 4.01 days, $t(50) = 1.922$, $P = 0.06$; 1st male eclosion: 47.7 ± 9.60 days vs 89.93 ± 36.33, $t(15.473) = 4.394$, $P < 0.001$; 1st gyne eclosion: 65.2 ± 9.78 days vs 106.3 ± 17.61, $t(27) = 7.681$, $P < 0.001$), had fewer workers in their first brood cohort (3.2 ± 1.12 vs 6.1 ± 2.21, $t(40.494) = 6.041$, $P < 0.001$), and produced fewer worker (13.8 ± 6.94 vs 136.7 ± 48.72, $t(27.138) = 12.964$, $P < 0.001$), male (14.4 ± 9.18 vs 43.9 ± 24.99, $t(16.853) = 4.352$, $P < 0.001$), and gyne (3.6 ± 2.17 vs 38.8 ± 17.08, $t(14.484) = 7.906$, $P < 0.001$) offspring (Fig. 1e). To determine if differences in colony size and productivity relate to foundress size, we randomly selected a subset of both inseminated workers and gynes that were used solely to measure anatomical features (Fig. 2a). Workers were, unsurprisingly for this species, significantly lighter than gynes (0.217 g vs. 0.736 g, $t(30.741) = 14.54$, $P < 0.001$), had smaller spermathecae (0.23 mm vs. 0.33 mm, $t(58) = 17.493$, $P < 0.001$), and had fewer sperm in their spermathecae (17,416 vs. 50,474, $t(43.843) = 22.519$, $P < 0.001$; Fig. 2b). However, given the number of sperm recorded, it is unlikely that spermatheca capacity limited colony size.

### Gene expression in inseminated workers and queens

To determine whether the expression of queen-like traits in inseminated workers shares the same molecular basis as those underlying true queen reproduction, we investigated gene expression in the reproductive tissues (spermatheca, vagina, and median oviduct) by comparing artificially inseminated workers and queens, which received sperm, with control workers and queens that did not receive the insemination treatment. First, through principal component analysis, we identified clear gene expression differences in the reproductive tract of workers vs. queens (PC1: 73%), followed closely by insemination treatment (PC2: 13%), which separated inseminated and control bees (Fig. 3a–c). Despite these tissue-wide differences in expression, both workers and queens shared conserved transcriptional responses to insemination with the differential expression of 193 genes (Fig. 3d–f) (Likelihood Ratio Test: False discovery rate (FDR) adjusted $P$ value < 0.05; $n = 165$ elevated DEGs; $n = 28$ reduced DEGs) in both inseminated workers and queens compared to control bees. A comparison of genome-wide $\log_2$ fold changes between inseminated and control bees for each morphological group (worker, queen) also identified a significant positive correlation (Pearson's correlation coefficient $R = 0.47$, $P < 2e\text{-}16$) providing additional support for conserved molecular responses to insemination in both workers and queens (Fig. 3g). In addition, a rank-based Gene Ontology (GO) term enrichment analysis identified differentially expressed genes for workers and queens as significantly enriched (Kolmogorov-Smirnov test: $P < 0.05$) for 47 and 84 GO terms for workers and queens, respectively, of which 32 terms were found in each morphotype-specific analysis, highlighting additional conserved responses to insemination. This suggests that either the same genes are used in response to insemination or similar non-targets are affected in each caste. Enriched GO terms in both workers and queens were associated with a range of diverse physiological processes, including reproduction, circadian rhythm, chemoreception, immunity, and mating behavior (Supplementary Data 8). Such enriched terms were also associated with gene co-expression networks significantly positively correlated with insemination ($R > = 0.67$, $p < 1e\text{-}05$; Supplementary Figure 12), which may represent genes associated with the organismal responses of females to mating, which in queens have been previously shown to involve reductions in receptivity to remate, as well as elevated immune potential[16,17].

### What enables the expression of mating behavior in workers?

Despite this functional anatomy and its molecular underpinnings, and the queen-like colony-rearing behavior displayed by artificially inseminated bumble bee workers, worker lifetime unmatedness may still

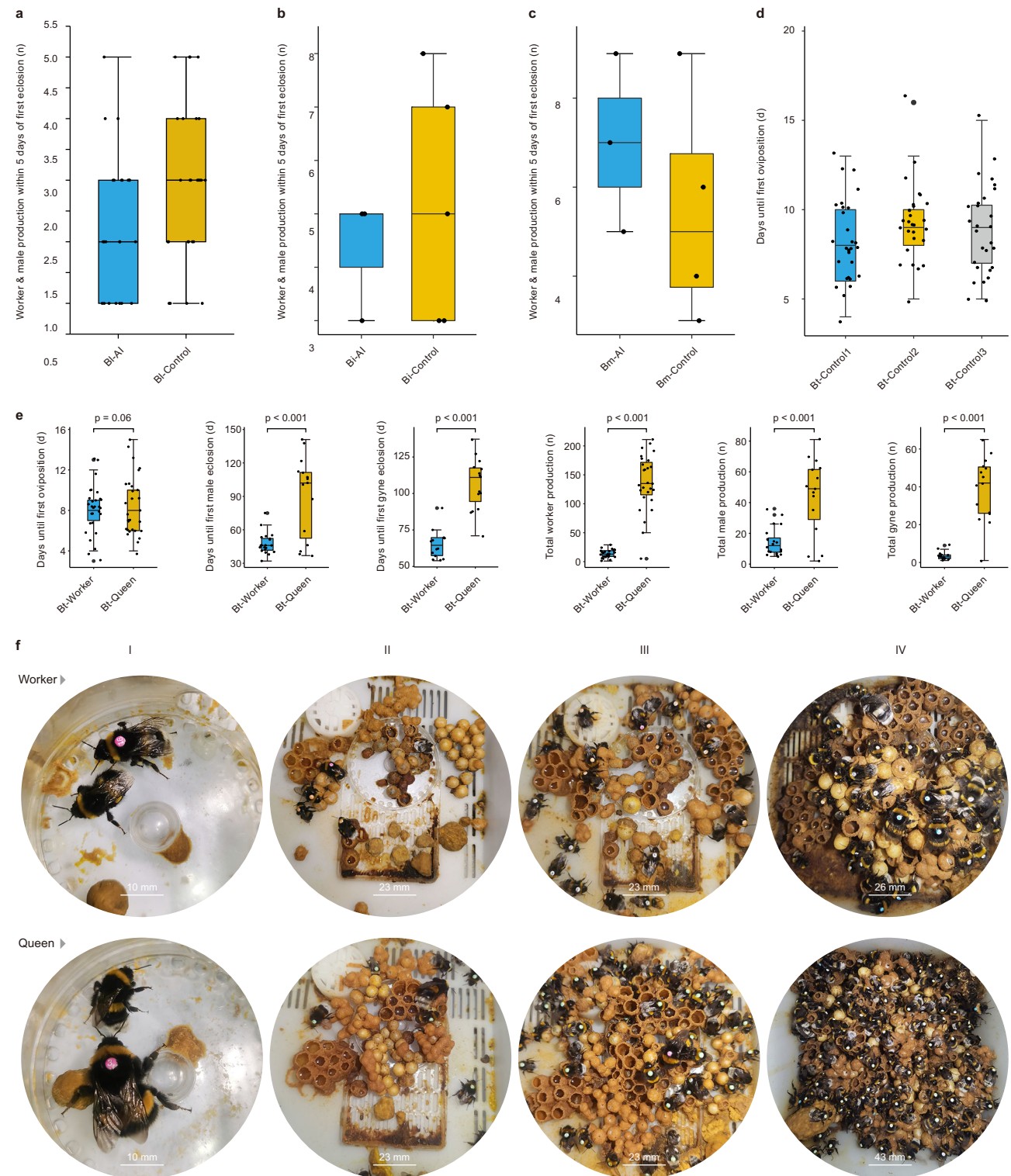

have evolved by selection against the ability to mate. In social insects, which exhibit a remarkable reproductive division of labor between queens and workers, worker reproduction is often determined by social cues[18]. To test whether bumble bee workers can mate, we first investigated if social isolation was associated with mating success in workers. For the purposes of all experiments outlined here and below, we collected newly eclosed *B. terrestris* callow workers from pupae obtained from queen-right colonies (here, queen-right colonies are defined as comprising of a queen and 30-40 workers without the presence of gyne or male larvae). Workers kept individually for seven

days were more likely to mate than workers that spent the same period of time in their natal colonies (31/90 vs 0/90 mated, respectively; Fisher's exact test *P* < 0.00001, Fig. 4a, b), providing the first evidence that social context post-eclosion influences mating potential. This is in sharp contrast to gynes, which do not leave the nest until ~seven days post-eclosion but are still able to mate (58/99, Fig. 4b). To determine if such patterns were species-specific, we repeated this experiment with two additional bumble bee species identifying that *B. lantschouensis* workers (12/45) and *B. ignitus* workers (9/45) can successfully mate after isolation from the natal social environment (Fig. 4b).

**Fig. 1 | Egg-laying and colony development for workers and queens in the artificial insemination experiments.** Worker and male production within five days of first eclosion by artificially inseminated (AI)(blue boxes) and control workers (orange boxes)(initial $N = 30$ in both groups for *B. lantschouensis* and *B. ignitus*, $N = 20$ in both groups for *B. montivagus*; figures are based on successful egg-layers, sample sizes shown below) **a** *B. lantschouensis* (AI = 23, Control = 26) **b** *B. ignitus* (AI = 3, Control = 5), and **c** *B. montivagus* (AI = 3, Control = 3). **d** Date of first oviposition by *B. terrestris* workers from treatments Control 1 (*Bombus terrestris* workers without artificial insemination, blue boxes, $N = 30$), Control 2 (*B. terrestris* workers with the artificial insemination procedure only, orange box, $N = 30$), and Control 3 (*B. terrestris* workers with artificial insemination who received the sperm diluent, grey box, $N = 30$). **e** Comparison of colony development between artificially inseminated *B. terrestris* workers (blue boxes) and queens (orange boxes)(initial $N = 30$ in both groups, sample sizes for individual figures shown below). *P* values were determined by two-sided *t*-tests. The date of first oviposition (Worker = 29,

Queen = 29), the eclosion date of the first batch of workers (Worker = 25, Queen = 27), the eclosion date of the first batch of males (Worker = 20, Queen = 15), the eclosion date of the first batch of new queens (Worker = 14, Queen = 15), the number of the first batch of workers (within 5 days of first eclosion)(Worker = 25, Queen = 27), the total number of workers per colony (Worker = 25, Queen = 27), the total number of males per colony (Worker = 20, Queen = 15), the total number of new queens per colony (Worker = 14, Queen = 15). Figures **a**–**e** show raw data and summary box plots where the box plots consist of the box denoting the interquartile range (IQR), bound by the 25th and 75th percentiles, the median line shown within the box, and the whiskers representing the rest of the data distribution with outliers denoted by points greater than ±1.5 x IQR. **f** Colonies produced by workers and queens went through the same colony stages: I: 1st brood; II: 2nd brood cells constructed on top of 1st brood pupae; III: Continuous worker brood production; and IV: Eventual switch to haploid egg-laying, including onset of competition phase.

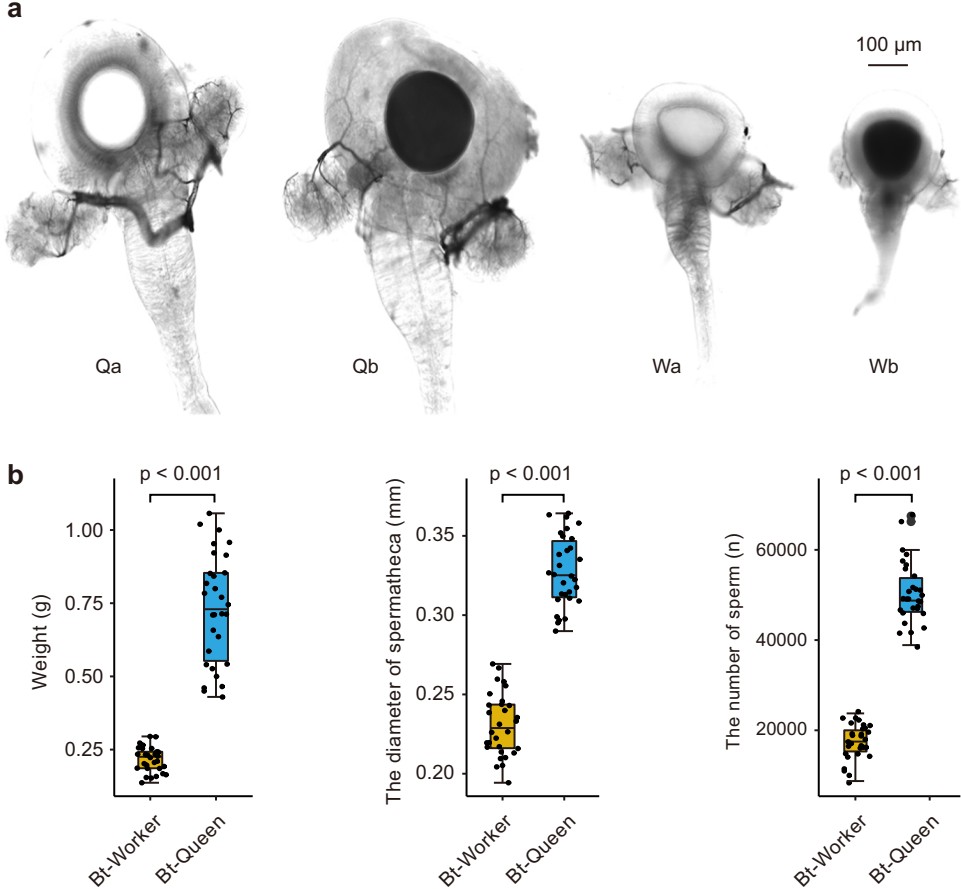

**Fig. 2 | The spermathecae, anatomy, and sperm number of artificially inseminated workers and queens in *Bombus terrestris*. a** Spermathecae of queens (Q) and workers (W) before (a) and after (b) artificial insemination. The spermathecae were photographed using a Leica TCS SP8 laser scanning confocal microscope with 20x magnification. **b** Queens (blue boxes, $N = 29$) had longer wings, were heavier, had larger spermathecae, and stored more sperm after insemination, compared to

workers (orange boxes, $N = 30$). Figure **b** shows raw data and summary box plots. Box plots consist of the box denoting the interquartile range (IQR), bound by the 25th and 75th percentiles, the median line shown within the box, and the whiskers representing the rest of the data distribution with outliers denoted by points greater than ±1.5 x IQR. *P* values were determined by two-sided *t*-tests.

To test what social factors might repress this mating ability, we first asked how worker age contributes to their likelihood of mating. We provided individual *B. terrestris* workers (removed as callows and maintained in individual boxes) the opportunity to mate at the age of 3, 4, 5, 6, 7, 8, 9, or 10 days old. Only three-day-old workers failed to mate (0/90, Fig. 4c), while five-day-old workers showed the highest mating propensity (49/96, Fig. 4c), which matches the onset of optimal mating age in bumble bee queens[19]. The inhibition of ovarian development and haploid egg-laying in worker bumble bees within colonies is mechanistically driven by physical contact with queens[20,21]. We,

therefore, investigated whether social control of worker mating is also driven by being in physical contact with the egg-laying queen[20]. For five days, we placed callow workers either in individual boxes ("isolated workers"), or individually in a box with an egg-laying queen, with this latter group subdivided into two treatments whereby bees were either allowed full physical interaction ("queen-contactable workers") or were separated by a wire mesh ("queen-separated workers"). Only workers that were physically separated from queens were able to mate (isolated workers = 41/90, queen-contactable workers = 0/90, queen-separated workers = 29/90; $G(2) = 58.58$, $P < 0.001$, Fig. 4d), indicating

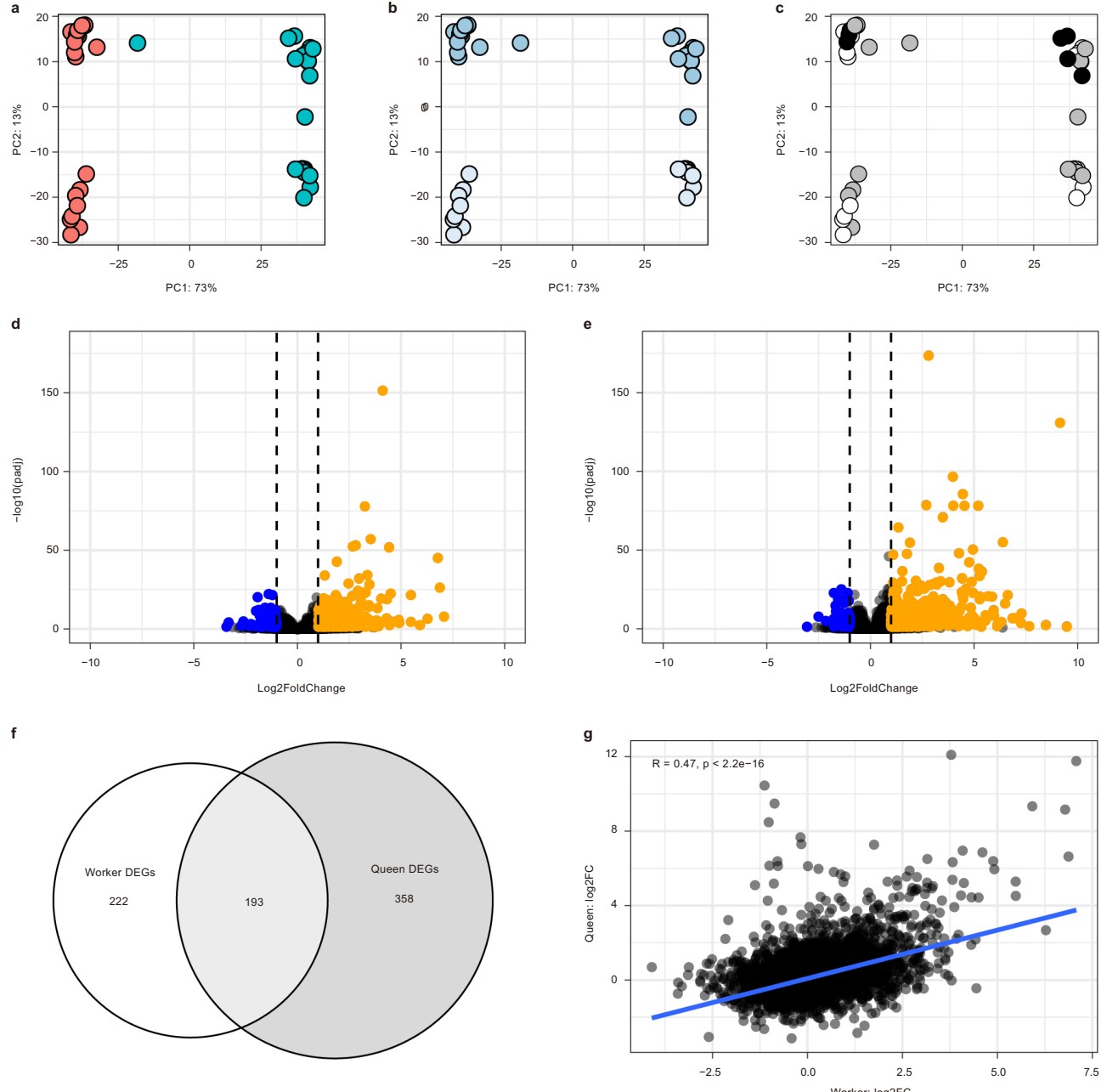

**Fig. 3 | Conserved transcriptional responses in *Bombus terrestris* workers and queens to insemination. a–c** Scatterplots displaying first and second principal components from a PCA performed on variance-stabilization transformed gene level counts revealing caste-specific gene expression profiles in gene expression profiles of reproductive tissues, including spermatheca, vagina and median oviduct, between: (**a**) the castes (red = queen; turquoise = worker); and (**b**) insemination status (blue = unfertilized (control) bees; pale blue = inseminated bees). In comparison, no clear separation of samples was identified based on days post-insemination (**c**: black dot = two days post-insemination; grey dot = four days post-insemination; and white dot = eight days post-insemination). **d, e** Differential expression analysis revealed similarities between queens and workers in terms of

response to insemination. For both (**d**) workers and (**e**) queens, insemination resulted in general patterns of elevated gene expression (Likelihood Ratio Test: Benjamini-Hochberg adjusted $P$ (padj) <0.05) compared to control bees (orange dots = elevated differentially expressed genes (DEGs) compared to control; blue dots = reduced DEGs compared to control; black dashed vertical lines indicate $log_2$ fold change (log2FC) thresholds for elevated (log2FC = 1) and reduced (log2FC = −1) gene expression. **f, g** Euler plot displaying overlap in DEGs shared by both queens and workers in response to artificial insemination while correlation analysis (Pearson's correlation coefficient) revealed a significant positive correlation between log2FC values assigned to genes affected by insemination.

that physical interactions with queens were necessary and sufficient to completely inhibit worker mating potential, although further research is needed to determine whether exposure to queen pheromones alone may inhibit workers mating.

In bumble bees, worker competition and policing can also inhibit the production of haploid eggs[18], therefore, we tested whether the presence of sister workers can also inhibit mating. We kept callow

workers either in a box on their own, with two workers of the same age, with two callow workers (which were replaced every 24 h with newly eclosed callows), or with two ovipositing workers for five days. Only workers kept on their own (41/89, Fig. 4e), with workers of the same age (32/90, Fig. 4e), or with callow workers (37/90, Fig. 4e) were able to mate when exposed to males while none of the workers (0/90, Fig. 4e) kept with ovipositing workers mated ($G(3) = 66.45$, $P < 0.001$). This

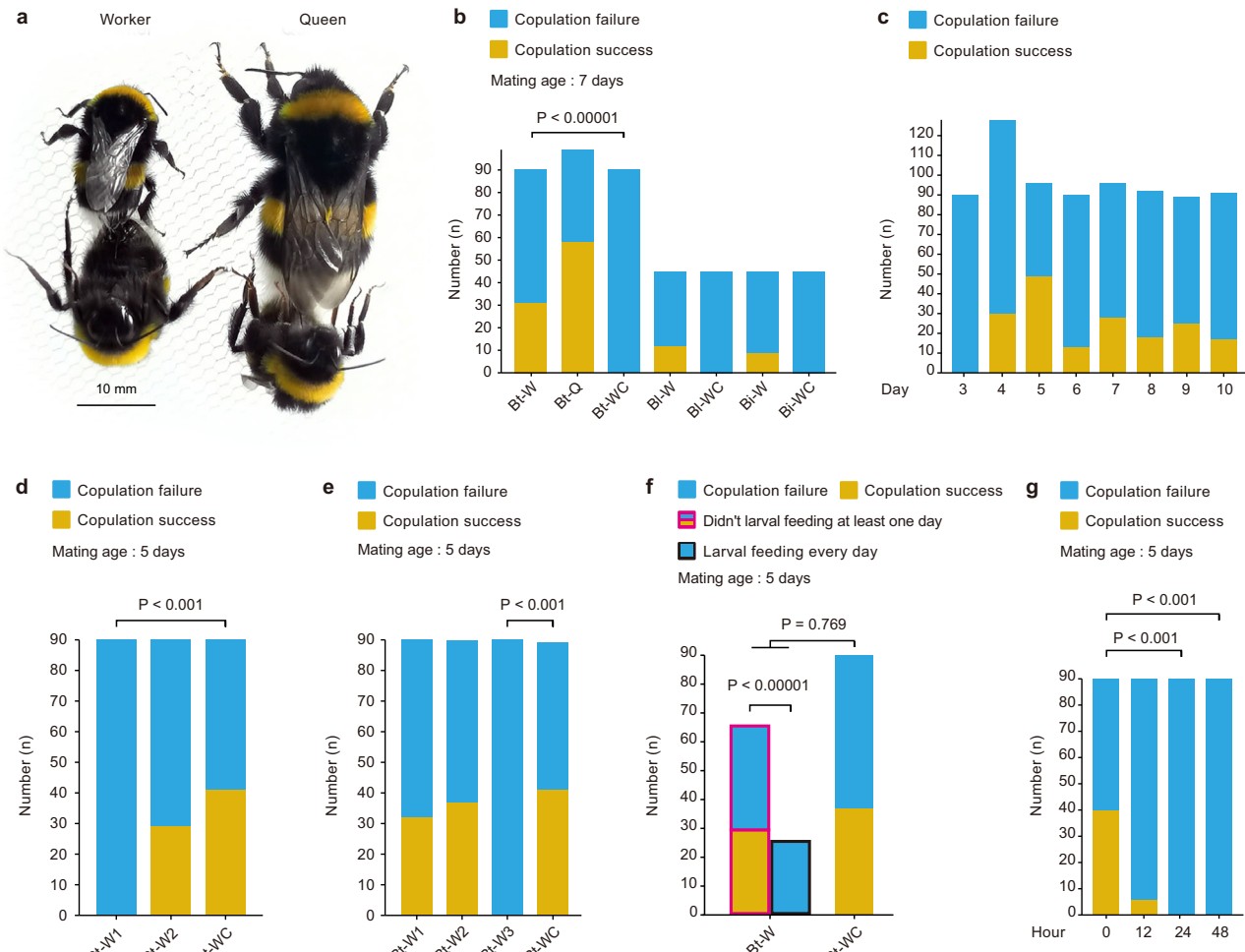

**Fig. 4 | Mating experiment and size comparison of mated workers and queens.**
**a** A *B. terrestris* worker (left) and queen (right) mating with males. **b** Isolation from the colony environment for seven days post-eclosion enabled worker mating: prior to mating trials at day seven, Bt-W = each *B. terrestris* callow worker was kept in one box (*N* = 90); Bt-Q = one *B. terrestris* gyne tagged as a callow was randomly selected from each colony (*N* = 99); Bt-WC = one *B. terrestris* worker tagged as a callow and returned to the colony was randomly selected from each colony (*N* = 90); Bl-W = *B. lantschouensis* callow workers were kept in a single box (*N* = 45); Bl-WC = one *B. lantschouensis* worker tagged as a callow and returned to the colony was randomly selected from each colony (*N* = 45); Bi-W = *B. ignitus* callow workers were kept in individual boxes for seven days (*N* = 45); and Bi-WC = one *B. ignitus* worker tagged as a callow and returned to the colony was randomly selected from each colony (*N* = 45). **c** *B. terrestris* worker age influences mating success (days 3–10, *N* = 90, 128, 96, 90, 96, 92, 89, 91, respectively). **d** Physical contact with queens inhibits *B.*
*terrestris* worker mating: Queen-contactable workers = each callow kept in a box with an egg-laying queen, enabling physical contact; Queen-separated workers = physical contact prevented by a metal mesh (ϕ 1 mm); Isolated workers = callows kept individually. **e** Worker presence inhibits mating ability: Bt-W1 = three callows collected on the same day kept in one box; Bt-W2 = one callow kept with two tagged additional callows, repeatedly replaced by new callows every 24 h; Bt-W3 = each callow kept with two tagged egg-laying workers; and Bt-WC = each callow kept alone. **f** Larval feeding behavior inhibits the mating ability of workers: Larval exposure = each callow kept with three larvae in one box; and Solitary = each callow kept alone. **g** Time spent in a social environment post–eclosion influences mating success. Data in **b** and **f** were analyzed using Fisher's test. Data in **d**, **e** and **g** were analyzed using G-tests. All mating experiments were repeated three times, and as patterns were consistent, data were combined for analysis.

result shows that physical contact with ovipositing workers is also sufficient to inhibit mating behavior. Consequently, mating inhibition in the colony context is also likely driven by exposure to ovipositing sisters.

Another factor that could potentially inhibit worker mating is the presence of brood, especially as we observed that contrary to accepted wisdom, callows within 24 h post-eclosion will feed larvae (Supplementary Movie 1). To test whether brood exposure inhibits mating, we kept newly-eclosed callow workers either with three larvae ("larval exposure"), or on their own ("solitary"), for five days. Upon first observation, no treatment inhibited mating (larval exposure: 29/90, solitary: 37/90; Fisher's exact test, *P* = 0.769, Fig. 4f). However, in the larval exposure group only a subset of workers (25/90, Fig. 4f) demonstrated continuous larval feeding behavior across the experiment, while the remaining workers largely ignored the larvae and

actively tried to escape confinement or spatially separated themselves from the larvae. None of the workers that exhibited continuous larval feeding mated (0/25, Fig. 4f), while significantly more workers that ignored the larvae mated (29/65; Fig. 4f, Fisher's exact test: *P* < 0.00001). Overall, only those workers who are not continuously engaged in worker-associated activities within the nest are likely to mate.

Finally, to determine the potential timeframe of social exposure that inhibits worker mating, we tagged (individually marked) callow workers, and then returned them to their natal colony for 0 (i.e., no return), 12, 24, or 48 h. After their designated duration of exposure, the tagged workers were removed again and kept individually in boxes until they were five days old. Only exposure to the natal colony for at least 24 h completely inhibited mating (0 h: 40/90, 12 h: 6/90, 24 hours: 0/90, and 48 h: 0/90, *G*(3) = 85.85, *P* < 0.001, Fig. 4g),

demonstrating that callows have a limited timeframe within which to escape the social environment before their mating ability is completely lost.

## Does queen-loss trigger worker mating?

Taken together, these results suggest that the ability of workers to mate may have been maintained by selection as a reproductive strategy that maximizes worker fitness in the case of queen loss[22]. To test whether bumble bee queen loss triggered natural worker mating, we conducted a semi-field experiment and found that workers were able to mate three days after the experimental removal of the natal queen. However, most of the successfully mated workers died within 24 h (11 worker bees mated in 7/15 colonies), with only two larger workers surviving. While causes of mortality remain unknown, it may be associated with physical incompatibilities between workers and males. While surviving mated workers returned to their natal colony and laid eggs, it was not possible to track reliably whether these eggs developed into female offspring. Consequently, we collected one worker who survived mating and returned to her natal colony, transferred her to a wooden box with worker pupae and workers from the natal colony, and placed them under semi-field conditions. New eggs appeared over the following night, and the first gyne eclosion was recorded after 41 days, indicating successful egg-laying and reproduction by the mated worker. While it remains unclear what determines which workers can mate in de-queened colonies, casual observations suggest that workers driven off the brood by unmated ovipositing sisters may escape the social factors previously identified (see above) to inhibit mating.

## Discussion

Although our experiments suggest that the likelihood of worker mating in a given de-queened colony is low, under natural conditions early queen death is likely to be a common occurrence (and is frequently observed during laboratory rearing of colonies; personal observation). Similarly, while most bumble bee males are produced at the end of the season, early queen death will also lead to the onset of worker competition and the production of early males[23]. Workers are attractive to males[24] and our results show that worker copulation may result from this attraction. Together, this suggests that low levels of worker mating in the wild may act as a selection pressure to maintain a functional spermatheca in bumble bee workers, and perhaps also in vespid wasps and certain ant subfamilies. However, identifying the frequency of such mating and subsequent worker reproduction in the wild is difficult. Finding and excavating sufficient wild colonies is challenging, due to the cryptic nature of bumble bee nests[25], although genetic analyzes of offspring from such nests may be able to distinguish between queen and worker diploid offspring. In its invasive range in Japan, *B. terrestris* colonies had anomalous patterns of genetic diversity among workers, which were interpreted as resulting from queen takeover or drifting;[26] re-analysis of these data in light of our results is warranted. In addition, bivoltinism has been reported in some annual bumble bees[27], often based on observations of two worker population peaks in the field. Our results suggest that such bivoltinism might sometimes result from worker mating, in addition to mated queens bypassing diapause. Finally, workers of an atypical perennial bumble bee[28] might be able to mate and reproduce in between the death of the colony queen and the re-queening of the colony[29].

In conclusion, our results indicate that the lifetime unmatedness of workers, in its strictest sense, is neither a pre-requisite for nor a necessary expectation of superorganisms. Bumble bee colonies, which have queens who mate once at the beginning of their life, and pre-imaginal determination of queen and worker morphotypes, are clearly superorganisms as defined by both early and recent definitions[1–3]. Our results suggest that workers will remain unmated under queen-right conditions, as expected by current theory[2,3], but that workers may

mate and rear colonies in response to queen-loss. Transcriptomic analyzes suggest that such workers are activating a gene regulatory network that is normally found in queens[3]. We suggest that there may be a difference between annual and perennial superorganisms, where the ecology of annual superorganisms selects for the retention of worker mating, and the ability to express a queen-like gene regulatory network, as a back-up strategy. Further studies investigating this in annual vespid wasps would be valuable. While the bulk of our experiments used commercial colonies of *B. terrestris*, there is no a priori reason to suggest that this would impact our conclusions. Firstly, where parallel experiments were conducted, results from commercial *B. terrestris* mirrored those from the three wild species we tested. Second, commercial conditions select for faster colony growth, larger colony size, and more queen production[30], and only queens are mated in commercial facilities, all of which are more likely to select against reproductive traits in workers. If our results can be extrapolated to other taxa, they may explain the apparent paradox of workers retaining intact spermatheca that apparently never receive sperm in superorganisms such as bumble bees, yellowjackets, and certain ant subfamilies. Finally, the ability of workers to mate and produce colonies provides an exciting new tool to combat global declines in bumble bees[31–35]. Artificial insemination of workers from rare and declining species could be used to generate large numbers of reproductive females for release back into the wild, reinforcing, and reintroducing bumble bee populations to habitats where they are essential pollinators.

## Methods

### Sampling bumble bees for insemination trials

We collected wild queens of *Bombus lantschouensis* and *B. ignitus* from Gansu province (E: 106.10, N: 34.26), China in April and May 2019, and reared them to produce colonies under laboratory conditions in environmentally-controlled rooms (temperature 28 °C ± 1 °C, relative humidity 60% ± 5%, in the dark). For *B. montivagus*, workers and males were collected from the field (E:103.89, N:24.78). Commercial colonies of *B. terrestris* were obtained from a commercial supplier based at the Institute of Apicultural Research, Chinese Academy of Agricultural Sciences.

All four species show strong queen-worker dimorphism (*B. terrestris*[36], for all four species, see Supplementary Fig. 1), which facilitated our size-based selection of workers and queens for experiments.

### Assessment of bumble bee worker spermathecae functionality

To assess the functionality of bumble bee worker spermathecae, we collected 60 callow workers of *B. lantschouensis*, three from each of 20 different queen-right colonies (here, and throughout, we define queen-right colonies as comprising a queen and 30-40 workers without gyne or male larvae) and 30 males were similarly collected from 10 different colonies. Each worker was kept individually in a small plastic box (6 cm*12 cm*8 cm) in environmentally-controlled rooms with 50% sucrose solution (W:W) and oilseed rape pollen provided *ad libitum*, prior to being allocated blindly to one of two treatment groups (see below). We artificially inseminated 30 workers, according to the method of Baer & Schmid-Hempel[15]. Each insemination was performed using a bespoke artificial insemination instrument for bumble bees (VE-AIIOQB-H1.0-A, Shanghai Suosheng Biotechnology Co., Ltd.,) and took place on 30ᵗʰ December, 2019. Post-insemination, to stimulate workers to lay eggs, each test individual was kept with two callow workers. The remaining 30 control workers were also each kept with two callow workers. These workers were removed after test workers began laying eggs to avoid affecting experimental results. To further confirm the reproductive function of the spermathecae of bumble bee workers, we repeated this experiment with two additional species: *B. ignitus* (again, 60 workers from 20 different queen-right colonies, and

30 males from an additional 10 colonies; $n = 30$ artificially inseminated workers, $n = 30$ control workers); and *B. montivagus* (workers and males collected in the field; $n = 20$ artificially inseminated workers, $n = 20$ control workers). More specifically, for *B. ignitus*, we inseminated workers in groups of six, which we then placed together as one group and reared in a small plastic box. In addition, we set up groups of six workers that did not receive the insemination treatment to act as controls. In total, we produced five groups for each treatment. For *B. montivagus*, we constructed four insemination and four control groups consisting of five workers each. All groups were monitored to determine whether the eggs laid developed into female or male offspring, as bumble bee workers are generally believed to only lay unfertilized haploid eggs, which develop into male offspring. For this, we recorded the number of the first batch of offspring workers (within five days of first eclosion) of artificially inseminated workers groups and the number of the first batch of offspring males (within five days of first eclosion) of control worker groups.

### Can bumble bee workers act as functional reproductive queens?

We used the buff-tailed or earth bumble bee, *B. terrestris* as a model system to test whether workers can act as functional queens. We collected 120 callow workers from 100 different queen-right colonies, 30 gynes from 30 different colonies, and 60 males from a further 30 colonies on 13[th] June, 2020. After collection, each bee was kept individually in a small plastic box as described above. At five days post-collection, 30 bees from each group were put into plastic bags (10 cm × 8 cm), from which the air was gently squeezed before carbon dioxide ($CO_2$ 99%) was added and maintained for 10 min to induce narcosis. Narcotized bees were individually weighed and tagged, with body length, body width, wing length, and wing width for each bee measured using a camera and the Bee Morphometer (MV-U210-V1.01-A). Measured bees were then returned to individual small plastic boxes prior to experimental treatments, which were performed at seven days post-eclosion. Each experimental group consisted of 30 bees and included: (i) an artificial insemination worker group, where workers were artificially inseminated with diluent and semen; (ii) a diluent-only insemination worker group containing workers that were handled in the same way as artificially inseminated workers but received only diluent and not semen; (iii) an empty insemination worker group containing workers that were handled in the same way as artificially inseminated workers but received neither diluent or semen; (iv) a worker control group containing bees that received no insemination; and (v) a queen artificial insemination group, which similar to the artificially inseminated workers, were artificially inseminated with both diluent and semen. In the insemination groups, each worker and queen were inseminated with 0.1 μl diluted semen from a single male while in the worker insemination diluent group, each worker was inseminated with 0.1 μl diluent only. After treatment, each bee was reared individually in a small plastic box, as described above. As outlined in the section above, to stimulate bees to lay eggs, in every treatment group two callow workers were provided to each bee, and removed after bees began laying eggs to avoid affecting experimental results. Incipient colonies were transferred to individual larger plastic boxes (20 cm × 20 cm × 14 cm) after the first batch of bees eclosed. All bees were observed every day and the following data were recorded: the date of first oviposition, the eclosion date of the first offspring (worker or male), the total number of offspring produced in the first batch (workers or males, within five days of first eclosion), the eclosion date of the first batch of gynes, as well as the total number of workers, males, and gynes produced per colony.

### Size and sperm content of spermathecae in workers and queens

We collected 30 callow gynes from 30 different colonies and 30 callow workers from 30 different queen-right colonies. Bees were kept individually as described above. At five days post-eclosion, each bee was anesthetized with carbon dioxide, tagged, and its body size and weight measured. At seven days post-eclosion, each worker and gyne was inseminated with 0.1 μl diluted semen from a single male. At 24 hours post-insemination, each bee was anesthetized with carbon dioxide, euthanized, and fixed on a wax disc with insect needles. We dissected along the tergum of the abdomen and the gut was removed carefully with tweezers under a x25 light microscope. The spermatheca was exposed and a drop of saline solution was quickly applied before the diameter of the spermatheca was then photographed and measured using the Bee Morphometer (MV-U210-V1.01-A) under a light microscope with x40 magnification. After measuring the diameter, each spermatheca was gently removed using tweezers and transferred to an individual Eppendorf tube containing 20 μl saline solution. The spermathecae were then punctured with forceps and stirred clockwise to ensure the spermatozoa had dispersed evenly into the solution. Sperm counts were then performed using a hemocytometer.

### Confirmation of worker offspring by microsatellite analysis

We performed microsatellite analysis to confirm that female offspring produced by artificially inseminated workers were diploid and shared the same father. We collected 30 queens, 124 workers, and 75 males from 12 different colonies to test and select microsatellite primers for further parentage identification of artificially inseminated workers and their offspring. Five artificially inseminated worker-produced colonies (Supplementary Data 3) were then used to identify the genetic relationship between workers and their offspring.

**Sample preparation and DNA extraction.** For DNA extraction, we sampled the thoracic muscles from each individual, which were first dissected from anaesthetized living bees using clean scissors. Internal thoracic muscles were used to avoid risk of contamination associated with the use of external components (e.g., tissue in contact with the exoskeleton) or internal tissues (e.g., the digestive tract). DNA was then extracted and purified using the Wizard® Genomic DNA Purification Kit (Promega, A1120) according to the manufacturer's instructions, with DNA suspended in 30 μL nuclease-free water. The concentration and quality of extracted DNA were assessed using a Qubit fluorometer (Invitrogen, Carlsbad, CA, USA) and 2% agarose gel electrophoresis, respectively. Extracted DNA was stored at −20 °C until further processing.

**Screening of microsatellite primers and PCR amplification.** For the purpose of assigning parentage, eight pairs of microsatellite primers were selected from the literature[37–39] (Supplementary Data 1). The microsatellite primers were synthesized (Berry Genomics Beijing Co., Ltd) with FAM and HEX fluorescent dyes for PCR amplification. The PCR reaction per sample comprised of 10 μl PCR Master Mix (TaKaRa, Dalian, China), forward and reverse primers, and template DNA at 0.8 μl each, and 7.6 μl ddH$_2$O to create a final reaction volume of 20 μl. Each PCR consisted of an initial 94 °C incubation step for 5 min, followed by 35 cycles of incubation at 94 °C for 1 min, 60 °C for 1 min, and 72 °C for 1 min, with a final extension step of 72 °C for 10 min. The amplified fragment sizes of the resulting PCR products were analyzed by capillary electrophoresis on an ABI-3730 sequencer (Applied Biosystems) with genotyping performed using GeneMarker v.2.2.0.

**Microsatellite polymorphism and parentage assignment analysis.** We analyzed the genotypic data using Cervus 3.0. For each microsatellite, we performed calculations for the observed heterozygosity (Hobs), expected heterozygosity (Hexp), Hardy Weinberg equilibrium (HW) and polymorphic information content (PIC). All eight microsatellite pairs had PIC values greater than 50%, making them appropriate to use for analyzes of relatedness within worker-produced colonies. For the determination of the genetic relationship between mothers and daughters, we then calculated single paternity exclusion

probabilities and cumulative exclusion probabilities (PCE), which allows for the removal of candidate parents based on genotypic mismatches but also accounting for the probability of typing error. Then in our parentage analysis with real data, the most likely candidate parent with a LOD (logarithm of the odds) score exceeding the critical LOD for 95% confidence can be assigned parentage with 95% confidence[40]. The estimated LOD value was used to evaluate the credibility of the most likely candidate parent.

### Gene expression in *B. terrestris* workers and queens

To determine whether artificially inseminated workers and queens undergo similar changes in gene expression, we conducted targeted transcriptomic analyzes of the reproductive organs (spermatheca, vagina, and median oviduct), as well as the brains, fat bodies, and ovaries of artificially inseminated and control workers and queens.

**Sample collection of reproductive tissue.** We collected callow workers ($n = 400$ worker bees) and assigned them randomly to one of five treatment groups: Control non-inseminated group: (i) two-day old control workers ($n = 80$); (ii) four-day-old control workers ($n = 80$); and (iii) eight-day-old control workers ($n = 80$); artificial insemination treatment groups: (iv) four day old artificially inseminated workers ($n = 80$) and (v) eight day old artificially inseminated workers ($n = 80$). For the artificially inseminated workers, insemination occurred 24 hours prior to sample collection, meaning that four and eight-day-old artificially inseminated workers were artificially inseminated at three and seven days post-eclosion, respectively. For sample collection, bees were anaesthetized, and tissues were dissected fresh. For each treatment and timepoint, we produced four biological replicates, each of which was comprised of tissues from 20 workers that were each collected from different colonies (Supplementary Data 7). As was done for the workers, gynes ($n = 200$) were allocated into one of the five treatment groups ($n = 40$ bees per treatment group), similar to those described above. The timing of artificial insemination and sample collection were as described above for workers. Again, as described for the workers, we produced four biological replicates for each treatment and timepoint, but these replicates were made by mixing 10 gynes samples collected from 10 different colonies (Supplementary Data 7). The different number of samples contributing to each biological replicate for workers and queens reflected the minimum amount of material required for sequencing, due to differences in spermatheca size between queens and workers. All samples were stored in Trizol and then frozen at −80 °C prior to RNA extraction.

**Sample collection for brains, fat bodies, and ovaries.** To determine if insemination in workers and queens affects other organs linked to reproduction, including behavior, we sampled brains, fat bodies and ovaries from workers and queens across three stages of ovarian development: stage I, the presence of immature ovaries with a thread-like appearance; stage II, the presence of nutritive cells (i.e., nurse cells) larger than the egg cell; and stage IV, the presence of mature eggs, observed after oviposition (Supplementary Fig. 2). For workers, at each stage of ovarian development, we sampled bees from four treatment groups: control workers ($n = 20$); artificially inseminated workers ($n = 20$); workers that received only diluent ($n = 20$); and workers that received neither diluent nor semen ($n = 20$). For workers, samples for each stage of ovarian development were obtained by collecting bees at three (stage I), four (stage II), and seven days (stage IV) post-eclosion. This gave a total of 80 bees for each stage and 240 bees in total across all three stages. For queens, there were two groups: control queens ($n = 20$); and artificially inseminated queens ($n = 20$). The same representative stages of ovarian development were obtained for queens by sampling of individuals at four (stage I), five (stage II), and seven days (stage IV) post-eclosion. For all bees, brain, fat bodies and ovaries were collected 24 hours after treatment for stages I and II, and 24 hours

within oviposition for bees for stage IV. For each treatment and stage of ovarian development per caste per tissue, each of which consisted of 20 bees, we created four replicates for sequencing by pooling five individuals (Supplementary Data 7). After pooling, all samples were stored in Trizol and then frozen at −80 °C prior to RNA extraction.

**RNA isolation, library preparation and sequencing.** From each sample, total RNA was extracted using a Trizol-based method, according to the manufacturer's instructions (Thermo Fisher Scientific, Carlsbad, CA, USA). RNA purity and quantity were evaluated using a NanoDrop 2000 spectrophotometer (Thermo Scientific, USA). RNA integrity was further assessed using an Agilent 2100 Bioanalyzer (Agilent Technologies, Santa Clara, CA, USA). Libraries were then constructed using the TruSeq Stranded mRNA LT Sample Prep Kit (Illumina, San Diego, CA, USA), according to the manufacturer's instructions. Libraries were individually barcoded, multiplexed, before sequencing (150 bp paired-end (PE)) and base calling was performed on an Illumina Novaseq 6000 by OE Biotech Co., Ltd. (Shanghai, China). Sequencing resulted in the generation of approximately 48.9 million raw reads on average per sample.

**Quality assessment and differential gene expression analysis.** Quality assessment of the raw sequencing data was performed using FastQC (v.0.11.9). Filtering of raw sequences was performed using fastp[41] (v.0.23.0) for the reproductive tissues while using Trimmomatic[42] (v.0.39) was used for the brains, fat bodies, and ovaries, to remove low quality reads and adapters, resulting in the retention of approximately 48 M (24 M PE) clean reads per sample. Filtered sequences were next aligned against the latest *B. terrestris* reference genome assembly available from Ensembl Metazoa (Bter_1.0[43]) using STAR[44] (v.2.7.4a), including the parameter '−quantMode GeneCounts' to produce gene-level counts. Differential gene expression analysis was performed using DESeq2[45] (v.1.26.0) with gene-level counts first loaded into a DESeq2 object. Gene-level counts were next filtered from this object to remove low or non-expressed genes (genes with less than a total of 10 reads across all samples). To determine similarities and differences among individuals and treatment groups in terms of expression profiles, hierarchical clustering and principal component analyzes were performed using counts following variance-stabilization transformation (VST), which was also implemented in DESeq2. For differential expression analysis, tissues and castes were analyzed independently with significantly differentially expressed genes (False Discovery Rate adjusted $P < 0.05$) determined using likelihood ratio tests implemented in DESeq2. Extended documentation of results can be found in Supplementary Information.

**Gene Ontology term enrichment analysis.** Given the depauperate nature of Gene Ontology (GO) terms assigned to most non-model organisms, we obtained and assigned the GO terms of the model organism *Drosophila melanogaster* to their *B. terrestris* homologs using resources available via Ensembl Metazoa BioMarts[46]. GO term enrichment analysis for differentially expressed genes was performed using topGO[47] (v.2.38.1; algorithm = "classic", node size = 20) for the implementation of rank-based Kolmogorov-Smirnov (KS) tests. For KS tests, we ranked all genes based on log₂ fold change to incorporate direction of expression into the analysis. Significantly enriched terms ($P < 0.05$) were visualized using a combination of the ggplot2 (v.3.3.6) and ggpubr (v.0.4.0) R packages. The present analysis used modified versions of scripts previously published by Colgan et al. [48].

**Weighted gene co-expression network analysis.** To better understand transcriptional responses in the reproductive organs of both queens and workers to insemination, we performed a weighed gene co-expression network analysis (WGCNA) with the WGCNA R package[49] (v. 1.71). Correlation matrices were first constructed and a

soft threshold power (β) was determined to generate adjacency matrices. We determined β using a measure of $R^2$ scale-free topology model fit. Using these soft power thresholds, adjacency matrices were generated and then converted into a topological overlap matrix (TOM) and a subsequent topological dissimilarity matrix (1-TOM) was generated. Using these values, we performed hierarchical clustering of genes, which allowed for the determination of gene modules (minClusterSize = 30; deepSplit = 2). Gene modules containing eigengenes that were highly correlated were merged (cutHeight = 0.25) resulting in the generation of consensus modules (each designated by an individual color). We then calculated correlation matrices between module eigengenes and traits of interest (i.e., caste, and treatment group). Furthermore, we also identified genes with high gene significance and module membership providing candidate genes of interest.

### What determines whether workers can mate?

To test whether bumble bee workers can mate, we first tested whether social isolation is associated with worker mating in three bumble bee species (*B. terrestris, B. lantschouensis* and *B. ignitus*), and then examined what factors might repress this ability using *B. terrestris*, as a model system to conduct a series of experiments. For each experiment, worker pupae were always collected from queen-right colonies. For each mating trial, the ratio of workers or gynes to males was 1:2, and the environmental conditions were as follows: the temperature was constant ($25\,°C \pm 1\,°C$) and the sizes of the mating cages were $50\,cm \times 50\,cm \times 50\,cm$. All mating trials were repeated three times.

### Does isolation from the colony enable mating?

For the purposes of understanding if social isolation affects the ability of bumble bees to mate, we collected 400 *B. terrestris* worker pupae from 100 different queen-right colonies, and 200 *B. terrestris* gyne pupae from 50 different colonies. We then kept these pupae in an incubator until eclosion (temperature $29\,°C \pm 1\,°C$, relative humidity $60\% \pm 5\%$). From these, we collected a total of 120 newly-eclosed callow workers. We transferred thirty callows immediately into individual small plastic boxes (see above). The remaining 90 callows were tagged and returned to 30 queen-right colonies, with a total of three callow workers returned to each colony. We also collected 90 newly-eclosed gynes from the population of gyne pupae, which were tagged and immediately placed in 30 queen-right colonies, with a maximum of three gynes returned per colony. For the mating experiment, after seven days, the isolated workers, and one tagged worker and one tagged gyne randomly selected from each colony, were given the opportunity to mate with males (Supplementary Fig. 3). To understand if the effect of social isolation on worker mating is conserved across other bumble bee species, we collected 150 *B. lantschouensis* worker pupae from 15 different queen-right colonies and 150 *B. ignitus* worker pupae from 15 different queen-right colonies. For each species, we then collected a total of 45 newly-eclosed callow workers, respectively. We immediately transferred 15 callows of each species into individual small plastic boxes (see above). The remaining 30 callows of each species were tagged and returned to 15 queen-right colonies of each species, with a total of two callow workers returned to each colony. Similar to as outlined above, for the mating experiment, after seven days, the isolated workers, and one tagged worker from each colony were given the opportunity to mate with males (Supplementary Fig. 3).

### Does worker age influence mating success?

We next asked how worker age contributes to their likelihood of mating. For this experiment, we collected 600 worker pupae from 150 different queen-right colonies and incubated them as outlined above. From this number, we used 210 newly-eclosed callows sampled on the same day, which were kept individually (one bee per box) prior to mating trials. To determine how age may influence mating, we randomly chose callows and exposed them to mating trials at days 3, 4, 5,

6, 7, 8, 9, and 10 post-eclosion. For each age group, we tested 30 workers (Supplementary Fig. 4) and recorded the number of observed mating events.

### Does exposure to queens inhibit worker mating?

Previous research has shown that workers are inhibited from reproducing when in contact with the queen[20]. To further determine whether social control of mating in workers is driven by queen contact, 300 worker pupae were removed from 100 different queen-right colonies and incubated as above to allow for maturation and subsequent eclosion. From these, 90 callow workers were randomly allocated equally across three groups: (i) callows kept individually (one bee per box); (ii) each callow was kept in a plastic box ($8\,cm \times 8\,cm \times 8\,cm$) with one queen that had laid eggs (i.e., egg-laying queen), allowing physical contact between worker and queen; and (iii) each callow was kept in a plastic box ($8\,cm \times 8\,cm \times 8\,cm$) with metal mesh (φ 1 mm) used to separate them from an egg-laying queen and preventing physical contact. Workers were kept in each treatment for five days before being exposed to a mating trial (Supplementary Fig. 5).

### Does exposure to other workers inhibit worker mating?

In bumble bees, worker competition and policing can limit the production and laying of haploid eggs[18]. Therefore, we tested whether the presence of sister workers inhibits worker mating. We collected 700 worker pupae from 300 different queen-right colonies as a source of callows. The control treatment consisted of 30 callows kept individually in small plastic boxes, and then exposed to a mating trial five days post-eclosion. To test whether the presence of same-aged workers affects the likelihood of a worker mating, we allocated 90 callows into 30 groups of three bees, kept in small plastic boxes as above, and after five days we randomly picked one worker per group and exposed it to a mating trial. To test whether the presence of younger worker bees inhibits worker mating, we collected 30 worker callows upon eclosion ("test worker") and placed them into individual small plastic boxes. We added a further two tagged, newly eclosed callows and repeatedly replaced these additional workers every 24 h. Each test worker was exposed to a mating trial at five days post-eclosion. To test whether egg-laying bees inhibit worker mating, we collected 30 newly eclosed callows ("test worker") and placed them in individual small plastic boxes, to which we added two tagged, egg-laying workers from colonies that had reached the "competition point", the natural colony stage when workers compete with the queen for reproductive output. All test workers were exposed to a mating trial at five days post-eclosion (Supplementary Fig. 6).

### Does feeding behavior affect the mating ability of workers?

An additional factor that could inhibit worker mating is the presence of brood. Larvae are reliant on adult workers for food and release signals to communicate with them[50]. The physical and chemical presence of larvae may lead to newly eclosed workers exhibiting characteristic worker behaviors, such as nursing and foraging, which may inhibit mating. To test this, we collected 200 worker pupae from 50 different queen-right colonies and incubated them as outlined above. From this number, we sampled 60 callows that eclosed on the same day and allocated them randomly to one of two treatment groups: (i) 30 callows were kept individually with three larvae and monitored via video recording for larval feeding behavior; or (ii) the last 30 callows were kept individually on their own without larvae. All test bees were provided the opportunity to mate at five days post-eclosion (Supplementary Fig. 7).

### Temporal dynamics of social exposure and worker mating

The duration of time spent in the colony post-eclosion may also influence the likelihood and ability of a worker to mate. To determine if there is an optimal timeframe for successful mating, we collected 500 worker pupae from 100 different queen-right colonies and incubated them as outlined above. From this number, we used 360 callow

workers that eclosed as adults on the same day. We divided these callow workers into four groups ($n = 90$ callows per group), with individuals in each group differing in the amount of time they spent in their natal colony post-eclosion. Upon eclosion, we assigned callows to one of the following groups and tagged them with a color representing the time spent in their natal colony: red = 0 h (i.e., no time spent in natal colony); yellow = 12 h; white = 24 h; and green = 48 h. Post-tagging, we returned callows to 30 natal queen-right colonies (three callows of each color in each colony) for their respective duration of exposure, after which we randomly selected and transferred one callow of each color from each colony to an individual small plastic box. The bees were kept in these small plastic boxes until they were five days post-eclosion, at which point they were exposed to a mating trial (Supplementary Fig. 8).

### Does queen-loss enable worker mating under semi-field conditions?

To test whether bumble bee queen loss triggered worker mating, we obtained 10 *B. terrestris* queen-right colonies and 10 male-producing colonies (i.e., without queen but with both adult worker and male bees present). For each colony, we collected and transferred all workers and the natal queen to a 1000 ml conical glass flask, and then moved the entire nest, including brood, to large wooden boxes (33.5 cm × 24 cm × 17 cm; Supplementary Fig. 9a). After the nest transfer, for each queen-right colony, we tagged each bee, and then tagged newly eclosed callows for four days leading up to the start of the experiment, on each day callows were tagged with a distinct color to record the day of eclosion (Supplementary Fig. 9b). After tagging, all bees were returned to the larger wooden box containing their respective colony. For the male-producing colonies, we did not tag workers but removed newly eclosed worker callows daily. Next, for the queen-right colonies, we removed the natal queen from each of the five colonies ("treatment group"), while the queen was not removed from the remaining five colonies ("control group"). We next placed 10 experimental mating cages (150 cm × 150 cm × 150 cm, Supplementary Fig. 9a) outdoors in the shade with each cage attached to three boxes (a worker-producing treatment colony, a male-producing colony, and an empty box; Supplementary Fig. 9a). We provided each mating cage with a 50% sugar water (W:W) solution, which was positioned in the middle of each cage (Supplementary Fig. 9a). We also provided pollen directly to the source colonies. We performed daily observations from 7:00 until 17:00 to observe and record the number of mating events involving worker bees. Mated workers were collected and tagged with a red mark on their thorax and followed to see if they flew back to their natal colony or to the empty box. If the mated bee flew back to the natal colony, we removed all the egg packages and L1 and L2 larval packages that were constructed on pupae using tweezers and observed whether mated workers laid eggs that developed into female offspring. This experiment was repeated three times. In order to track reliably whether these eggs laid by mated workers developed into female offspring, we collected a worker that survived mating, transferred her to a large wooden box with pupae and workers from her natal colony, and placed them under semi-field conditions to allow for egg-laying and colony development. We then recorded the time of the new egg appearance, the eclosion time of gynes, and the total number of gynes produced.

### Statistical analyzes

We used R v4.1.1 and SPSS for analyzes. Colony development, spermathecae, anatomy, and sperm number of artificial insemination worker and queen data were analyzed using *t*-tests. Data from mating experiments were analyzed using Fisher's test or G-tests.

### Reporting summary

Further information on research design is available in the Nature Portfolio Reporting Summary linked to this article.

## Data availability

The sequence data generated in this study have been deposited in the NCBI Sequence Read Archive database under accession code PRJNA868857. Other experimental data are available as Supplementary Data. Source data are provided with this paper.

## Code availability

Scripts used for the transcriptomic analysis are available at: https://github.com/Joscolgan/bombus_mated_worker_analysis.

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

## Acknowledgements

This study was supported by The Agricultural Science and Technology Innovation Program (CAAS–ASTIP-2016–IAR, J.L), the Chinese National Natural Science Foundation (No. 31572338, J.L). We also appreciate help from the Shanghai OE Biotech Co., Ltd, which assisted with the sequencing of RNA-seq libraries on their Illumina Novaseq 6000. We also thank Berry Genomics Beijing Co., Ltd for assistance with and use of their ABI Sequencer, which was used for barcoding and the microsatellite work. We also thank Ben Sadd from Illinois State University, Wenjun Peng, Yongjian Wang from State Key Laboratory of Resource Insects, Institute of Apicultural Research, Chinese Academy of Agricultural Science, Weiyu Yan from Jiangxi Agricultural University, Fucai Liang from Nanning Guangxi, Guanghui Zhao and Jiancai Li from Yunnan for collecting samples, as well as providing help and advice in the laboratory. We would particularly like to thank Koos (JJ) Boomsma and Guojie Zhang for their helpful discussions.

## Author contributions

Conceived and designed the experiments: M.Z., X.D., M.J.F.B., and J.L. performed the experiments: M.Z., Yulong Guo, Yueqin Guo, Zhengyi Zhang, F.L., Z.X., Y.L., L.W., J.X., Y.Q., J.Y., H.Y., X.L., J.G., J.L. Analyzed the data: M.Z., T.J.C., Yulong Guo, Zhengyi Zhang, Zhihao Zhang, F.Y., M.J.F.B., and J.L. Contributed reagents/materials/analysis tools: M.Z., T.J.C., Yulong Guo, Zhengyi Zhang, Y.Q., X.D., M.J.F.B., and J.L. Wrote the paper: M.Z., T.J.C., M.J.F.B., and J.L. All authors contributed to and approved the final version.

## Competing interests

The authors declare no competing interests.
