## [Peer Review File · Nature Communications]

Unexpected worker mating and colony-founding in a superorganismReviewers' Comments:

Reviewer #1:

Remarks to the Author:

The major transitions, in which previously independently reproducing units combine to form a novel reproducing unit at the higher level, represent key steps in the history of life, and their study therefore represents a fundamental topic in evolutionary biology. Eusocial insects with morphological differentiation between a reproducing caste (queens or kings) and a non-reproducing, helping caste (workers) represent a well-studied and prominent major transition. In the eusocial Hymenoptera, lifetime commitment of individuals to a worker role occurs in several lineages, including bumblebees. It is associated with loss of mating ability (although, via haplodiploidy, unmated workers can still produce male offspring from unfertilised eggs), since inability to mate successfully means that a worker can never become a foundress queen capable of producing its own worker offspring, so binding the worker to its worker role. Such lifetime commitment to unmatedness has therefore been recognised as a critical step in the occurrence of the major transition to eusociality.

Against this background, the current study shows that, contrary to what was generally understood, worker bumblebees are, like queens, capable of mating, storing sperm in a functional sperm receptacle, and then founding colonies based on their production of female worker offspring from fertilised eggs, with these colonies undergoing a colony cycle similar to that of queens. As the bumblebee lineage is thought from phylogenetic evidence to have achieved caste-differentiated eusociality tens of millions of years ago, the retention of mating ability and function to the extent demonstrated is surprising and unexpected. It is also conceptually important because (a) it shows that absolute loss of the ability to assume a fully queen-like role among workers is not an inevitable consequence of caste-differentiated eusociality, and (b) it suggests that the evolution of complete sterility of workers found in some lineages may occur in a series of steps widely separated over evolutionary time. The study is very thorough and of a high standard, employing several species and multiple, sophisticated methods. Artificial insemination of worker bees must be technically demanding, and the authors are to be commended for having thought to do this original experiment. The manuscript is overall well written and clearly presented. For these reasons, the study represents a substantial, novel contribution to our understanding of the major transition to eusociality and of major transitions in general, and should find a broad readership of interested researchers. I have a few major points to suggest as revisions to enhance the case presented and anticipate reader questions, followed by minor points to improve the effectiveness of the presentation.

Major points:

1. Conceptual background: To clarify its case (which I think is a strong one) to be of general relevance to the discipline, the ms could be more specific as to the concepts it addresses. This is partly covered in the introduction (lines 61-77) but, for example, at line 255, the ms states that the results are 'in contrast to previous theory', and it would benefit from being clearer as to exactly what theory or concept is being challenged.

2. Phylogenetic and morphological context: The phylogenetic context of the study species is considered at lines 87-89, but greater consideration of where the study species sit in the bumblebee subfamily phylogeny would be helpful. Related to this, bumblebee subfamilies vary in their level of queen-worker size dimorphism, and this has been linked to the division of species into so-called 'pollen storers' and 'pocket makers' (e.g. Michener CD. 1974. *The Social Behavior of the Bees*. Belknap Press, Cambridge MA; p. 98). So it would be useful to know which of these types the study species fall into, and/or their levels of queen-worker size dimorphism. More specifically, I may have missed them, but, if not, the criteria by which workers in the current study were designated as such could be more clearly stated, as it seems critical to be assured that the workers were not in some cases 'small queens' in species with low queen-worker size dimorphism.

3. Natural behaviour context: A question arising from the results concerns the extent to which, though it may be possible, worker mating and/or colony founding occur and are important in nature. Although little might be known of this, maybe because investigators have not looked for it, the ms would benefit from including further consideration of this issue, even if briefly. This is especially the case as, though workers mated with males in the semi-field experiment, the males were from colonies placed next to the colonies with the workers. It also appears that mating with males physically damaged many workers (lines 221-223). So workers may not always have opportunities to mate in the field, and those that do may mate subject to both costs and benefits. With regard to the general point, some possible aspects for consideration are:

(a) Commercial colonies v. colonies reared from wild-caught queens: The study used a mix of these two types of colony, so some consideration of the potential effect of commercial rearing on workers' mating behaviour might be useful.

(b) Evidence for worker mating in the field: Worker mating, to be successful in nature, would presumably have to fit with existing mating systems of bumblebees, which often involve females responding to a set of specific behaviours of males (patrolling, scent marking, etc.). Any evidence from the literature that workers respond like queens to such behaviours might therefore be informative (especially as it may be relatively rare for male-producing colonies to be physically close to colonies with potentially mating workers).

(c) Evidence for worker colony founding in the field: Likewise, any evidence from the literature for workers founding colonies in the field would be useful. Some annual bumblebees have been reported to show bivoltinism (e.g. Hart AF et al. 2021. *Apidologie* 52: 315), and this is sometimes inferred from observations of late peaks of worker numbers in the field. Conceivably, then, such peaks might be related to worker colony founding occurring in the field, and so could be worth discussion.

(d) Queen replacement: The ms suggests that worker mating is a 'back-up strategy' against early queen loss (line 217), but, more generally, in other bee species in which females adopting worker roles retain the ability to mate, this is used to replace/challenge queens as the main breeding female of the colony on a routine basis, e.g. in a queuing system (e.g. Vickruck JL, Richards MH. 2018. *Insectes Sociaux* 65: 619). It seems unlikely that bumblebee researchers have missed this occurring in bumblebees (although challenge by unmated reproductive workers is well evidenced), and it would seem to require in-nest mating or mating outside the nest followed by a return to the nest; but any information from the literature on this point, whether in the lab or field, would again be useful.

Minor points:

4. Line 31: 'Superorganism' remains a slightly contentious term, so maybe reword to something like: 'The emergence of caste-differentiated colonies, which have been termed 'superorganisms', in ...'

5. Line 39: Since evolution of other traits will also have occurred, maybe reword to 'despite 25-40 million years of eusocial evolution'.

6. Line 51: To match the sense, reword to 'maintain intact spermathecae yet apparently never mate'.

7. Line 63: The first author of ref. 1 (Koos Boomsma) has expanded the arguments in that article in his recent book, so this could also be cited here (Boomsma JJ. 2022. *Domains and Major Transitions of Social Evolution*. Oxford University Press, New York.).

8. Line 77: Replace 'system' with 'taxon' here?

9. Line 81: Replaced 'laid' with 'produced'.

10. Line 93: The 'While' at the start of this sentence implies a contrast is to follow, but it is not quite clear why the information in the two clauses is being contrasted.

11. Lines 96-97, 'Microsatellite analysis confirmed that these female offspring of each worker-produced colony were diploid and shared the same father': To make this completely clear, it should be added that, as the sense implies, the father sharing paternity in each case was the one whose sperm had been used for artificial insemination of the mother worker.

12. Line 149: Delete 'the' before 'worker'.

13. Line 150: Change 'worker' to 'workers'.

14. Line 157: The text seems to variously use 'hatching', 'emergence' and 'eclosion' to mean the same thing (eclosion), and it would be better if it used one term only, preferably the more accurate 'eclosion'.

15. Line 169-170, 'As bumble bee queens...': The main point being made is that, at the mechanistic level, unmated workers refrain from egg laying in the queen's presence if workers experience physical contact with the queen, and so the sentence could be reworded to state this a little more clearly.

16. Lines 174-177: It would be clearer if the terms used here ('isolated bees', 'queen-contactable bees' etc.) were used consistently throughout, whereas in the Figure 4d legend a different terminology is used ('Bt-W2' etc.). Plus, in 'isolated bees' etc., 'workers' would be a better term than 'bees', as being more specific.

17. Lines 197-198: Again, the terminology in the text ('larval exposure' etc.) is not applied consistently in the figure, and the ms would be clearer if it were.

18. Line 209: 'Tagged' could be defined at first use (i.e. here) as meaning 'individually marked', as this is more specific.

19. Lines 216-217, 'Taken together, these results suggest that the ability of workers to mate may have been selected for as a back-up reproductive strategy': In this sentence, '... may have been maintained by selection...' might be more accurate, plus the notion of a 'back-up' strategy requires some examination. It implies a sort of colony-level concern to prolong the colony's life, but it might be more accurate to rephrase as, in some contexts, and given the opportunity, worker mating might serve workers' fitness interests, i.e. as a form of queen replacement.

20. Lines 217-234: The semi-field experiment shows that workers from dequeened colonies can mate with males from other colonies. It also returned only one surviving mated worker that may have produced gynes; for this latter finding of the experiment there is therefore a small sample size and apparently some uncertainty, so it might be better to reconsider the space given to it in the main narrative and include it instead as supplementary information, plus reword the sentence about it in the Abstract. The results in Fig. 1 have already established the successful foundation of colonies by mated workers (in the laboratory).

21. Lines 238-240, 'Similarly, while most bumble bee males are produced at the end of the season, early queen death will also lead to the onset of worker competition and the production of early males': The effect of the timing of queen death on workers' production of males can be attested to with data, since Almond et al. (2019) showed that sequentially earlier removal of queens led to (unmated) workers producing a greater proportion of males (Almond EJ et al. 2019. *American Naturalist* 193: 256).

22. Line 242: Add 'a' before 'functional'.

23. Line 252: Replace 'studies' with 'workers'.

24. Lines 344, 347: The generic name could be abbreviated after first use.

25. Lines 352-357: More detail could be given on the artificial insemination methodology and the selection of workers and males. For example: clarify whether the instrument is a bespoke one for bumble bees or, which seems more likely, one manufactured for honey bee queen artificial insemination; state how many source colonies the males, treatment workers and control workers were drawn from; state if males were used from colonies different from those providing the treatment workers, and likewise if treatment and control workers were from the same colonies as one another or different ones.

26. Line 400: Clarify if all worker types were provided with the two callow workers each (in this experiment and all similar ones).

27. Line 473: The sense suggests that this subheading and later subheadings should be at a lower level than the one at lines 467-468.

28. Line 486, 'Similar to the workers, we produced...': This construction (here and elsewhere) does not quite work grammatically; change to something like, 'As in the case of the workers, we produced...'.

29. Lines 556-570: The results of this analysis (weighted gene co-expression network analysis) are mentioned in the Supplementary Information (line 47-), but I couldn't find them referred to in the main text.

30. Line 636, 'We added an additional...': To avoid the repetition, maybe change to 'We added a further...'.

31. Figures:

(a) In general, it would help if all figure legends stated the sample sizes (N workers, N colonies, or both, as appropriate), where this information is not on the figure itself (e.g. it appears not be on Figure 1 or in its legend). In addition, on the Y axis labels, the lack of a space between the label and the units is confusing, e.g. as in 'DFO(d)' on Fig. 1d; all such labels would be clearer with a space before the unit (in brackets).

(b) Figure 1: The Y axis labels for Figs 1a-c, e.g. 'TNFBW(M)5', appear to be composites of abbreviations defined in the legend, which is confusing: the labels should correspond exactly with the versions defined in the legend.

(c) Figure 3: Figs. 3a-c all appear to have the same points, but they are classified (and therefore coloured) differently; if so, it would be helpful to state this in the legend. At line 805, the legend defines yellow points in Fig. 3b, but the coloured points on this figure don't include any yellow ones.

(d) Figure 4: In the legend, add definitions of 'mated failure' and 'mated success' (terms in the figures): for example, whether these refer to copulation observed, sperm transferred, or some other criterion. At lines 826-827, clarify if the randomly selected worker was also callow. For Figs 4c and beyond, state the species represented in the panel. For Fig. 4f, define in the legend the terms 'not continuous larval feeding' and 'continuous larval feeding'.

32. Supplementary Information:

(a) Line 54: Reword to 'significance of the correlation'.

(b) Extended Data Fig. 1: State which species the images are from. In general, since the study uses several *Bombus* species, all figure and table legends should be explicit as to species.

(c) Extended Data Fig. 10: The legend refers to colours (line 275) but the figure panels show no colours, at least in the viewable pdf; this also applies to subsequent similar figures.

(d) Supplementary Tables: I may have overlooked it but a table of the microsatellite genotypes would be informative.

Andrew Bourke

Reviewer #2:

Remarks to the Author:

The worker caste of bumble bees retain a complete and thus presumably functional spermatheca. The main result of this paper is to demonstrate using artificial insemination on three *Bombus* species that the sperm storage organ is indeed functional. This result is interesting because it is apparently the first such demonstration, which was a surprise to me (Was it not known that bumble bee workers could potentially mate?). The authors complement this experimental result and observation with a full suite of ad hoc side experiments to show how this ancestral parental behaviour - sex- is associated with social (e.g., age, number and type of interactions) and genetic (e.g., alleles, gene expression) factors. The methodology seems sound and thorough. In my view, this paper represents a substantial amount of work that is well presented and likewise incorporates a lot of knowledge into the biology of bumble bees. The broad significance of the paper is, however, not clear and, from my reading, seems to be based on a slightly faulty narrative - namely, that the weakly eusocial breeding system of bumble bees somehow represents an 'evolutionary mystery' or paradox that has now been resolved. What, exactly, is the paradox?

Is it that workers have a complete spermatheca, despite being mostly non-reproductive? To me, the functional spermatheca makes a lot of evolutionary sense - namely, it is retained under direct selection for workers to mate and reproduce when opportunities arise, for example, when their queen dies. As the authors conclude, it is a back-up strategy. This does not seem a mystery or paradox to me; rather, it is reasonable and consistent with all of inclusive fitness theory that predicts conditional expression of direct versus indirect fitness strategies, at least for castes that are not yet locked-in to one or the other, as is the case for bumble bees and other weakly eusocial species. Anyway, that workers retain queen-like qualities, including in this case the ability to rear offspring, is normal for many social insect species, given that the queen and worker caste diverged from a common form.

Finally, the authors argue that their artificial-insemination result refutes an understanding that superorganismality cannot evolve without obligate worker un-matedness. This does not seem necessary. Superorganismality - i.e., eusociality - is a matter of degree, as measured by the level of worker reproductive altruism. Alas, un-matedness (and more generally, worker sterility) and eusociality co-evolve together, step by step. We don't expect superorganismality to suddenly appear upon the origin of obligate unmatedness, do we? If so, then I guess bumble bees were never superorganisms and the argument dissolves anyway. In total, despite the interesting success at coaxing workers to lay fertilized eggs and other interesting lab results, the author's narrative about resolving an evolutionary paradox or refuting theories about social evolution do to seem warranted.

Author Responses to Referee Comments (reviewer comments in blue, responses indented and in black)

Reviewer #1

The major transitions, in which previously independently reproducing units combine to form a novel reproducing unit at the higher level, represent key steps in the history of life, and their study therefore represents a fundamental topic in evolutionary biology. Eusocial insects with morphological differentiation between a reproducing caste (queens or kings) and a non-reproducing, helping caste (workers) represent a well-studied and prominent major transition. In the eusocial Hymenoptera, lifetime commitment of individuals to a worker role occurs in several lineages, including bumblebees. It is associated with loss of mating ability (although, via haplodiploidy, unmated workers can still produce male offspring from unfertilised eggs), since inability to mate successfully means that a worker can never become a foundress queen capable of producing its own worker offspring, so binding the worker to its worker role. Such lifetime commitment to unmatedness has therefore been recognised as a critical step in the occurrence of the major transition to eusociality.

Against this background, the current study shows that, contrary to what was generally understood, worker bumblebees are, like queens, capable of mating, storing sperm in a functional sperm receptacle, and then founding colonies based on their production of female worker offspring from fertilised eggs, with these colonies undergoing a colony cycle similar to that of queens. As the bumblebee lineage is thought from phylogenetic evidence to have achieved caste-differentiated eusociality tens of millions of years ago, the retention of mating ability and function to the extent demonstrated is surprising and unexpected. It is also conceptually important because (a) it shows that absolute loss of the ability to assume a fully queen-like role among workers is not an inevitable consequence of caste-differentiated eusociality, and (b) it suggests that the evolution of complete sterility of workers found in some lineages may occur in a series of steps widely separated over evolutionary time. The study is very thorough and of a high standard, employing several species and multiple, sophisticated methods. Artificial insemination of worker bees must be technically demanding, and the authors are to be commended for having thought to do this original experiment. The manuscript is overall well written and clearly presented. For these reasons, the study represents a substantial, novel contribution to our understanding of the major transition to eusociality and of major transitions in general, and should find a broad readership of interested researchers. I have a few major points to suggest as revisions to enhance the case presented and anticipate reader questions, followed by minor points to improve the effectiveness of the presentation.

Author Response 1.0: We would like to thank Professor Bourke for his appreciation of, and positive assessment of our manuscript.

Major points:

Reviewer Comment 1.1. Conceptual background: To clarify its case (which I think is a strong one) to be of general relevance to the discipline, the ms could be more specific as to the concepts it addresses. This is partly covered in the introduction (lines 61-77) but, for example, at line 255, the ms states that the results are 'in contrast to previous theory', and it would benefit from being clearer as to exactly what theory or concept is being challenged.

Author Response 1.1: We appreciate the opportunity to clarify this point. As the reviewer has acknowledged, our findings challenge the concept that a hallmark of superorganismality is "preimaginal determination of a life-time unmated worker caste that is typically morphologically distinct" by demonstrating that "bumblebee workers are, like queens, capable of mating, storing sperm in a functional sperm receptacle, and then founding colonies based on their production of female worker offspring from fertilised eggs, with these colonies undergoing a colony cycle similar to that of queens". As noted by the reviewer, we detail this in the Introduction (and have now gone into this in more detail - see response below). However, in line with this comment from the reviewer, and to improve clarity, we have now clearly explained the theory we state as being challenged in lines 272-273:

" , where lifetime unmatedness is expected of workers in superorganismal bumble bee colonies,"

Reviewer Comment 1.2. Phylogenetic and morphological context: The phylogenetic context of the study species is considered at lines 87-89, but greater consideration of where the study species sit in the bumblebee subfamily phylogeny would be helpful. Related to this, bumblebee subfamilies vary in their level of queen-worker size dimorphism, and this has been linked to the division of species into so-called 'pollen storers' and 'pocket makers' (e.g. Michener CD. 1974. The Social Behavior of the Bees. Belknap Press, Cambridge MA; p. 98). So it would be useful to know which of these types the study species fall into, and/or their levels of queen-worker size dimorphism. More specifically, I may have missed them, but, if not, the criteria by which workers in the current study were designated as such could be more clearly stated, as it seems critical to be assured that the workers were not in some cases 'small queens' in species with low queen-worker size dimorphism.

Author Response 1.2: We agree with the reviewer that the phylogenetic position of the species studied here is important, particularly to understanding how our results might generalise across the genus. Specifically, three of the four species included in the present study (*B. lantschouensis*, *B. ignitus*, and *B. terrestris*) belong to *Bombus sensu stricto*, a group estimated to have appeared ca. 15 million years ago (MYA) (Hines 2008, *Systematic Biology*). However, the evolutionary distance between the subgenera *Bombus sensu stricto* and *Megabombus*, which contains *B. montivagus*, is estimated at ~25 MYA (Hines 2008, *Systematic Biology*). This implies that either the most recent common ancestor of these two subgenera, which is relatively basal to the entire bumblebee phylogeny, had workers with the ability to mate, or it was lost and then subsequently regained secondarily in two distinct subgenera. The former is the most parsimonious explanation. We now give more detail on this in the manuscript in lines 88-100 (shown below with new text shown in yellow highlighting):

“To determine if such patterns were species-specific, we repeated this experiment with two additional bumble bee species, *B. ignitus* (which is in the same subgenus as *B. lantschouensis*, *Bombus sensu stricto*, with a common ancestor ~8MYA^{Error! Reference source not found.}) and *B. montivagus* (which is in the subgenus *Megabombus*, sharing a common ancestor with the subgenus *Bombus s.s.* ~25MYA^{Error! Reference source not found.}), which both responded similarly to the artificial insemination (AI) treatment by producing female offspring (Fig. 1b,c). As these species diverged from a common ancestor ~25 MYA^{Error! Reference source not found.}, the most parsimonious explanation of our findings suggests that such abilities and associated mechanisms may be conserved across all social bumble bees.

To understand if bumble bee workers can act as functional queens (i.e., rear a colony through the complete lifecycle), we used the buff-tailed or earth bumble bee, *B. terrestris* (also a member of the subgenus *Bombus s.s.*, which shares a common ancestor with *B. ignitus* ~6MYA^{Error! Reference source not found.})”

In relation to the second point about caste size dimorphism, *B. lantschouensis*, *B. ignitus*, and *B. terrestris* belong to *Bombus s. s.* a 'pollen-storer' clade, where queen-worker size dimorphism is evident (Williams et al 2008 *Apidologie*; for *B. terrestris* Alford 1975 and see Fig. 1 below, for *ignitus* and *lantschouensis* see Fig. 1 below). While *B. montivagus* belongs to the subgenus *Megabombus*, members of which are generally 'pocket makers' (Williams et al 2008 *Apidologie*), they also show distinct queen-worker size dimorphism (Fig. 1 below). Consequently, given that we selected bees for our experiments by size, concerns about potential confusion of small queens and large workers are unfounded. In addition, while this dimorphism facilitated our ability to select queens and workers correctly for our studies, for *B. terrestris* (as stated in the manuscript) we also sampled individuals from relatively young colonies (1 queen plus 40 workers) before the onset of the “competition point”, which was determined by an absence of worker competition (aggression, egg-eating, egg-laying). We have now revised our Methods and Supplementary Information sections to include this information and the new figure shown below, see lines 300-302 and Section A in the Supplementary Information:

“All four species show strong queen-worker dimorphism (*B. terrestris*³⁶, for all four species see Supplementary Figure 1), which facilitated our size-based selection of workers and queens for experiments.”

Supplementary Figure 1. Size dimorphism in bumblebee castes. A) Histograms displaying weight distributions, a proxy for body size, for complete colony populations of female bees taken from colonies of four bumblebee species; and B) Boxplots displaying weight measurements for workers and queens - where caste is designated based on the size distributions shown in A) above - again based on complete populations from colonies of four bumblebee species. For each type of plot, species are individually coloured: *Bombus montivagus* = grey, *B. ignitus* = orange, *B. lantschouensis* = purple, and *B. terrestris* = blue, with the species name also found in the grid header. For each species, we plot the dataset for a single representative colony indicated by the number found inside the brackets within the grid header while for three of the species, we plot a cumulative plot of all populations collected from all colonies for that species (n = 15).

Reviewer Comment 1.3. Natural behaviour context: A question arising from the results concerns the extent to which, though it may be possible, worker mating and/or colony founding occur and are important in nature. Although little might be known of this, maybe because investigators have not looked for it, the ms would benefit from including further consideration of this issue, even if briefly. This is especially the case as, though workers mated with males in the semi-field experiment, the males were from colonies placed next to the colonies with the workers. It also appears that mating with males physically damaged many workers (lines 221-223). So workers may not always have opportunities to mate in the field, and those that do may mate subject to both costs and benefits. With regard to the general point, some possible aspects for consideration are:

(a) Commercial colonies v. colonies reared from wild-caught queens: The study used a mix of these two types of colony, so some consideration of the potential effect of commercial rearing on workers' mating behaviour might be useful.

Author Response 1.3a: This is an interesting point. We used workers caught directly from the field, workers taken from colonies reared from wild-caught queens, and workers from commercial colonies for artificial insemination and colony-rearing. None of these groups showed any differences in their ability to be successfully inseminated and found colonies. All further experiments used only workers from commercial bees. While it is likely that commercial colonies are selected for a range of features, incl. faster growth, larger colony size, and more queen production (Gosterit & Baskar 2016 *Insectes Sociaux* 63:609], we can think of no *a priori* reason why worker mating behaviour would be selected for during domestication, as workers are never given the option to mate in commercial facilities (and each new generation of commercial colonies is derived from mated queens). Consequently, it seems most parsimonious that the worker mating behaviours we observed are intrinsic to wild bumblebees. We have added additional text to discuss this in lines 277-283 (pasted below):

“While the bulk of our experiments used commercial colonies of *B. terrestris*, there is no *a priori* reason to suggest that this would impact our conclusions. Firstly, where parallel experiments were conducted, results from commercial *B. terrestris* mirrored those from the three wild species we tested. Second, commercial conditions select for faster colony growth, larger colony size, and more queen production^{Error! Reference source not found.}, and only queens are mated in commercial facilities, all of which are more likely to select against reproductive traits in workers.”

(b) Evidence for worker mating in the field: Worker mating, to be successful in nature, would presumably have to fit with existing mating systems of bumblebees, which often involve females responding to a set of specific behaviours of males (patrolling, scent marking, etc.). Any evidence from the literature that workers respond like queens to such behaviours might therefore be informative (especially as it may be relatively rare for male-producing colonies to be physically close to colonies with potentially mating workers).

Author Response 1.3b: We agree with the reviewer that evidence for such worker behaviour would be great to be able to cite. Unfortunately, to the best of our knowledge, all that is known is that workers are chemically attractive to males (Gudrun et al 2006, already cited in our manuscript), and from our study that receptive workers will mate with males when given the opportunity. We would note, however, that bumble bee nest density can be high (e.g., ~50 nests/hectare, Cumber 1953) facilitating potential male/worker interactions. Interestingly, we also know surprisingly little about how queens respond to male behaviours in the field (reviewed by Alford 1975, with no work investigating this since to the best of our knowledge). We hope that our results will encourage future studies that examine these behaviours in detail in both workers and queens.

(c) Evidence for worker colony founding in the field: Likewise, any evidence from the literature for workers founding colonies in the field would be useful. Some annual bumblebees have been reported to show bivoltinism (e.g. Hart AF et al. 2021. *Apidologie* 52: 315), and this is sometimes inferred from observations of late peaks of worker numbers in the field. Conceivably, then, such peaks might be related to worker colony founding occurring in the field, and so could be worth discussion.

Author Response 1.3c: This is an excellent and intriguing suggestion, which had not occurred to us. As a result, we have added the following text in the Discussion section (lines 266-269):

“In addition, bivoltinism has been reported in some annual bumble bees²⁷, often based on observations of two worker population peaks in the field. Our results suggest that such bivoltinism might sometimes result from worker mating, in addition to mated queens bypassing diapause.”

We would like to add the parenthesis “{pers. comm. AFG Bourke)” to acknowledge and attribute the source of this idea, if that would be deemed appropriate by the reviewer and the journal?

(d) Queen replacement: The ms suggests that worker mating is a ‘back-up strategy’ against early queen loss (line 217), but, more generally, in other bee species in which females adopting worker roles retain the ability to mate, this is used to replace/challenge queens as the main breeding female of the colony on a routine basis, e.g. in a queuing system (e.g. Vickruck JL, Richards MH. 2018. *Insectes Sociaux* 65: 619). It seems unlikely that bumblebee researchers have missed this occurring in bumblebees (although challenge by unmated reproductive workers is well evidenced), and it would seem to require in-nest mating or mating outside the nest followed by a return to the nest; but any information from the literature on this point, whether in the lab or field, would again be useful.

Author Response 1.3d: This is an interesting perspective, but such behaviour has only been reported in non-superorganismal species (for example, the paper cited by the reviewer is on carpenter bees). We agree that it is unlikely bumblebee researchers have missed such a behaviour. To our knowledge, the only relevant study in the literature is by Matos & Garofalo (1995) where workers in queenless colonies of the perennial species *B. atratus* have been suggested to mate and lay eggs until new queens emerge and take over. Given that we had

already cited this where relevant in the discussion in our manuscript, and the unlikely nature of the suggested behaviour occurring in a superorganismal species, we have not made changes to the manuscript in this case. We hope that the reviewer agrees that this is an appropriate decision, but if not, we are happy to revisit this point.

Minor points:

Reviewer Comment 1.4. Line 31: 'Superorganism' remains a slightly contentious term, so maybe reword to something like: 'The emergence of caste-differentiated colonies, which have been termed 'superorganisms', in ...'

Author Response 1.4: We have reworded this sentence as suggested. See lines 32-33. Please note, to comply with journal guidelines we have rewritten the entire abstract, but we believe it responds to all the reviewer points here and below

Reviewer Comment 1.5. Line 39: Since evolution of other traits will also have occurred, maybe reword to 'despite 25-40 million years of eusocial evolution'.

Author Response 1.5: Due to the change in the abstract we were not able to change as suggested, but we believe that the new language removes the concern raised

Reviewer Comment 1.6. Line 51: To match the sense, reword to 'maintain intact spermathecae yet apparently never mate'.

Author Response 1.6: Unfortunately, due to the word limit for the abstract we were not able to include this text.

Reviewer Comment 1.7. Line 63: The first author of ref. 1 (Koos Boomsma) has expanded the arguments in that article in his recent book, so this could also be cited here (Boomsma JJ. 2022. Domains and Major Transitions of Social Evolution. Oxford University Press, New York.).

Author Response 1.7: Thank you, we have cited this reference wherever we refer to Boomsma, and have added it to the reference list as reference 3. See References, line 682-683.

Reviewer Comment 1.8. Line 77: Replace 'system' with 'taxon' here?

Author Response 1.8: We have changed "system" to "taxon". See line 66.

Reviewer Comment 1.9. Line 81: Replaced 'laid' with 'produced'.

Author Response 1.9: We have replaced "laid" with "produced". See line 84. Please note, we have also rewritten this last paragraph to bring it in line with journal requirements.

Reviewer Comment 1.10. Line 93: The 'While' at the start of this sentence implies a contrast is to follow, but it is not quite clear why the information in the two clauses is being contrasted.

Author Response 1.10: Thank you for noticing this. It was because we left out a word in the sentence ("only"). We have added this in, and the contrast is, we believe, clear now. See line 102.

Reviewer Comment 1.11. Lines 96-97, 'Microsatellite analysis confirmed that these female offspring of each worker-produced colony were diploid and shared the same father': To make this completely clear, it should be added that, as the sense implies, the father sharing paternity in each case was the one whose sperm had been used for artificial insemination of the mother worker.

Author Response 1.11: Thank you. We have added the text "fathered by the male whose sperm had been used for artificial insemination" to make this clear. See lines 105-106.

Reviewer Comment 1.12. Line 149: Delete 'the' before 'worker'.

Author Response 1.12: We have deleted "the" before "worker". See line 161.

Reviewer Comment 1.13. Line 150: Change 'worker' to 'workers'.

Author Response 1.13: We have changed "worker" to "workers". See line 162.

Reviewer Comment 1.14. Line 157: The text seems to variously use 'hatching', 'emergence' and 'eclosion' to mean the same thing (eclosion), and it would be better if it used one term only, preferably the more accurate 'eclosion'.

Author Response 1.14: We have changed “hatching” and “emergence” to “eclosion”, and “hatched” and “emerged” to “eclosed” throughout the entire manuscript. These changes are highlighted in yellow

Reviewer Comment 1.15. Line 169-170, ‘As bumble bee queens...’: The main point being made is that, at the mechanistic level, unmated workers refrain from egg laying in the queen’s presence if workers experience physical contact with the queen, and so the sentence could be reworded to state this a little more clearly.

Author Response 1.15: We have rephrased this sentence, which now reads:
“The inhibition of ovarian development and haploid egg laying in worker bumble bees within colonies is mechanistically driven by physical contact with queens”. See line 181-183.

Reviewer Comment 1.16. Lines 174-177: It would be clearer if the terms used here (‘isolated bees’, queen-contactable bees’ etc.) were used consistently throughout, whereas in the Figure 4d legend a different terminology is used (‘Bt-W2’ etc.). Plus, in ‘isolated bees’ etc., ‘workers’ would be a better term than ‘bees’, as being more specific.

Author Response 1.16: We have changed “bees” to “workers” throughout the whole manuscript where appropriate (highlighted in yellow). We have changed the x-axis labels for Fig 4d.

Reviewer Comment 1.17. Lines 197-198: Again, the terminology in the text (‘larval exposure’ etc.) is not applied consistently in the figure, and the ms would be clearer if it were.

Author Response 1.17: We have corrected this in the Figure legend, see lines 899-901, and in the x-axis labels of Fig 4f.

Reviewer Comment 1.18. Line 209: ‘Tagged’ could be defined at first use (i.e. here) as meaning ‘individually marked’, as this is more specific.

Author Response 1.18: We have defined “tagged” at first use as requested. See line 221.

Reviewer Comment 1.19. Lines 216-217, ‘Taken together, these results suggest that the ability of workers to mate may have been selected for as a back-up reproductive strategy’: In this sentence, ‘... may have been maintained by selection...’ might be more accurate, plus the notion of a ‘back-up’ strategy requires some examination. It implies a sort of colony-level concern to prolong the colony’s life, but it might be more accurate to rephrase as, in some contexts, and given the opportunity, worker mating might serve workers’ fitness interests, i.e. as a form of queen. replacement.

Author Response 1.19: Thank you, this is an excellent point. We have rephrased these sentences to make this clear. See lines 230-232.

Reviewer Comment 1.20. Lines 217-234: The semi-field experiment shows that workers from dequeened colonies can mate with males from other colonies. It also returned only one surviving mated worker that may have produced gynes; for this latter finding of the experiment there is therefore a small sample size and apparently some uncertainty, so it might be better to reconsider the space given to it in the main narrative and include it instead as supplementary information, plus reword the sentence about it in the Abstract. The results in Fig. 1 have already established the successful foundation of colonies by mated workers (in the laboratory).

Author Response 1.20: While we understand the point that the reviewer is making here, and we do recognise the low sample size, we believe that this result is actually an important proof of principle. We have reduced the text referring to it in the main narrative but retain it there because we believe it is an important addition to the dataset.

Reviewer Comment 1.21. Lines 238-240, ‘Similarly, while most bumble bee males are produced at the end of the season, early queen death will also lead to the onset of worker competition and the production of early males’: The effect of the timing of queen death on workers’ production of males can be attested to with data, since Almond et al. (2019) showed that sequentially earlier removal of queens led to (unmated) workers producing a greater proportion of males (Almond EJ et al. 2019. American Naturalist 193: 256).

Author Response 1.21: Thank you for bringing this paper to our attention. We have now added this reference in the discussion. See lines 254, 730-732.

Reviewer Comment 1.22. Line 242: Add ‘a’ before ‘functional’.

Author Response 1.22: We have added “a” before “functional”. See line 256.

Reviewer Comment 1.23. Line 252: Replace 'studies' with 'workers'.

Author Response 1.23: We have replaced "studies" with "workers". See line 269.

Reviewer Comment 1.24. Lines 344, 347: The generic name could be abbreviated after first use.

Author Response 1.24: Thank you we have abbreviated the generic name after first use throughout the Methods.

Reviewer Comment 1.25. Lines 352-357: More detail could be given on the artificial insemination methodology and the selection of workers and males. For example: clarify whether the instrument is a bespoke one for bumble bees or, which seems more likely, one manufactured for honey bee queen artificial insemination; state how many source colonies the males, treatment workers and control workers were drawn from; state if males were used from colonies different from those providing the treatment workers, and likewise if treatment and control workers were from the same colonies as one another or different ones.

Author Response 1.25: The instrument we used was bespoke-made for this work, and we have now stated this on lines 312-313. We have also clarified the colony origins of males and workers, see lines 304-308.

Reviewer Comment 1.26. Line 400: Clarify if all worker types were provided with the two callow workers each (in this experiment and all similar ones).

Author Response 1.26: We have clarified this, as requested. See line 315-316, 358.

Reviewer Comment 1.27. Line 473: The sense suggests that this subheading and later subheadings should be at a lower level than the one at lines 467-468.

Author Response 1.27: Thank you. We have re-formatted these to make them lower-level subheadings.

Reviewer Comment 1.28. Line 486, 'Similar to the workers, we produced...': This construction (here and elsewhere) does not quite work grammatically; change to something like, 'As in the case of the workers, we produced...'

Author Response 1.28: We have rephrased these sentences as suggested. See lines 442-443.

Reviewer Comment 1.29. Lines 556-570: The results of this analysis (weighted gene co-expression network analysis) are mentioned in the Supplementary Information (line 47-), but I couldn't find them referred to in the main text.

Author Response 1.29: We have now added information about the results of the WGCNA to the Results section (Lines 150-153):

"Such enriched terms were also associated with gene co-expression networks significantly positively correlated with insemination ($R \geq 0.67$, $p < 1e-05$; Supplementary Information; Supplementary Figure 12), which may represent genes associated with the organismal responses of females to mating, which in queens have been previously shown to involve reductions in receptivity to remate, as well as elevated immune potential^{Error! Reference source not found.}Error! Reference source not found."

Reviewer Comment 1.30. Line 636, 'We added an additional...': To avoid the repetition, maybe change to 'We added a further...'

Author Response 1.30: We have changed this as suggested. See line 595.

Reviewer Comment 1.31. Figures:

(a) In general, it would help if all figure legends stated the sample sizes (N workers, N colonies, or both, as appropriate), where this information is not on the figure itself (e.g. it appears not be on Figure 1 or in its legend). In addition, on the Y axis labels, the lack of a space between the label and the units is confusing, e.g. as in 'DFO(d)' on Fig. 1d; all such labels would be clearer with a space before the unit (in brackets).

Author Response 1.31a: We have added sample numbers and modified figure legends where appropriate. See lines 823-840.

(b) Figure 1: The Y axis labels for Figs 1a-c, e.g. 'TNFBW(M)5', appear to be composites of

abbreviations defined in the legend, which is confusing: the labels should correspond exactly with the versions defined in the legend.

Author Response 1.31b: Thank you for highlighting this issue. We have edited the figure legend to remove the acronyms and the x- and y-axis labels now reflect what is in the figure legend. See Fig 1 and Fig 1 legend lines 823-840

(c) Figure 3: Figs. 3a-c all appear to have the same points, but they are classified (and therefore coloured) differently; if so, it would be helpful to state this in the legend. At line 805, the legend defines yellow points in Fig. 3b, but the coloured points on this figure don't include any yellow ones.

Author Response 1.31c: We thank the reviewer for highlighting this issue and have revised the figure legend to correct it (Line 862-863).

(d) Figure 4: In the legend, add definitions of 'mated failure' and 'mated success' (terms in the figures): for example, whether these refer to copulation observed, sperm transferred, or some other criterion. At lines 826-827, clarify if the randomly selected worker was also callow. For Figs 4c and beyond, state the species represented in the panel. For Fig. 4f, define in the legend the terms 'not continuous larval feeding' and 'continuous larval feeding'.

Author Response 1.31d: We have made all these changes, and used the term 'copulation' in the figure to make it clear as to what we measured. We have re-described larval feeding in the figure itself to make it clear. To clarify, we tagged three callows and put them into one colony, then randomly selected one mature, tagged worker at 7 days from each colony before exposing them to a mating trial - see Figure 4, line 884, line 887, line 889.

Reviewer Comment 1.32. Supplementary Information:

(a) Line 54: Reword to 'significance of the correlation'.

Author Response 1.32a: We have reworded. See Supplementary Information line 70.

(b) Extended Data Fig. 1: State which species the images are from. In general, since the study uses several *Bombus* species, all figure and table legends should be explicit as to species.

Author Response 1.32b: We have explicitly described the species as requested. See highlighted text in the Supplementary Materials. See Supplementary Information line 218.

(c) Extended Data Fig. 10: The legend refers to colours (line 275) but the figure panels show no colours, at least in the viewable pdf; this also applies to subsequent similar figures.

Author Response 1.32c: We thank the reviewer for bringing this issue to our attention, which we have now corrected.

(d) Supplementary Tables: I may have overlooked it but a table of the microsatellite genotypes would be informative.

Author Response 1.32d: We have added a new supplementary table of microsatellite genotypes. See supplementary table 6.

Reviewer #2 (Remarks to the Author):

Reviewer Comment 2.0: The worker caste of bumble bees retain a complete and thus presumably functional spermatheca. The main result of this paper is to demonstrate using artificial insemination on three *Bombus* species that the sperm storage organ is indeed functional. This result is interesting because it is apparently the first such demonstration, which was a surprise to me (Was it not known that bumble bee workers could potentially mate?). The authors complement this experimental result and observation with a full suite of ad hoc side experiments to show how this ancestral parental behaviour - sex- is associated with social (e.g., age, number and type of interactions) and genetic (e.g., alleles, gene expression) factors. The methodology seems sound and thorough. In my view, this paper represents a substantial amount of work that is well presented and likewise incorporates a lot of knowledge into the biology of bumble bees. The broad significance of the paper is, however, not clear and, from my reading, seems to be based on a slightly faulty narrative - namely, that the weakly eusocial breeding system of bumble bees somehow represents an 'evolutionary mystery' or paradox that has now been resolved. What, exactly, is the paradox?

Author Response 2.0: We thank the reviewer for their constructive comments. To clarify, while demonstration of spermatheca functionality by artificial insemination is an important result of our study, we do not believe it stands alone as the main result. The aim of our research was to a) determine whether spermathecae in bumblebee workers were functional, b) whether workers could rear colonies, c) whether workers could mate with males, d) under what circumstances workers were enabled to express this behaviour, and e) understand some of the mechanisms underpinning worker-queen reproduction through gene expression. As such, and with respect, we do not view parts b)-e) of our research as 'ad hoc side experiments'. They are part of an integrated study, which we now make clear in the Introduction, see lines 67-77.

We are glad that the reviewer found the demonstration of functionality interesting. As cited in our manuscript, there was evidence from one study in a perennial bumblebee species that workers might be able to mate, but our study is the first to demonstrate this robustly and to explore the broader reasons around, and implications of it. We also appreciate that the reviewer found our methodology to be sound and thorough, and that our manuscript adds considerably to the body of knowledge.

We are sorry that the novelty and broad significance of our study did not come through clearly (although see Reviewer 1, Andrew Bourke, who explicitly recognised both the novelty and the importance of this study). Below, we hope we address this concern clearly. However, before doing so, we note that the reviewer refers to the breeding system of bumble bees as 'weakly eusocial'. We believe that this might underlie many of the points raised by the reviewer. While historically, bumble bees were often referred to as primitively or weakly eusocial, recent comprehensive work (as described and cited in our manuscript) in developing a sound theoretical and evolutionarily underpinned structure for social classification clearly placed them as belonging to a group of 'superorganismal' taxa (other members include ants, vespine wasps, and corbiculate bees more broadly) that possess a range of integrated traits. While we thought we had described this in depth in our manuscript, it clearly did not come across as clearly as it could have. We believe that the revisions we have made in response to Reviewer 1 (Bourke) make this much clearer (please see referenced new text above).

Finally, we address the paradox question below (Author Response 2.1), where it is further developed by the reviewer.

Reviewer Comment 2.1: Is it that workers have a complete spermatheca, despite being mostly non-reproductive? To me, the functional spermatheca makes a lot of evolutionary sense - namely, it is retained under direct selection for workers to mate and reproduce when opportunities arise, for example, when their queen dies. As the authors conclude, it is a back-up strategy. This does not seem a mystery or paradox to me; rather, it is reasonable and consistent with all of inclusive fitness theory that predicts conditional expression of direct versus indirect fitness strategies, at least for castes that are not yet locked-in to one or the other, as is the case for bumble bees and other weakly eusocial species. Anyway, that workers retain queen-like qualities, including in this case the ability to rear offspring, is normal for many social insect species, given that the queen and worker caste diverged from a common form.

Author Response 2.1: In essence, yes, the paradox we refer to is indeed the retention of an energetically-costly organ, the spermatheca, despite 35-40 MYA since the advent of superorganismality. While the reviewer suggests that this is not surprising, given inclusive fitness theory, this relies upon the categorisation of bumble bees as weakly eusocial where castes are not yet locked in. However, bumblebees are not weakly eusocial, and their castes are locked in. To elaborate, and as described in the manuscript, bumblebees belong to a group of taxa (ants, vespine wasps, and corbiculate bees) that are categorised as 'superorganismal' (Boomsma 2022). The life-time unmatedness of helper castes, such as workers, is one of the major transitions linked to the origin of superorganismality (Boomsma 2022). With respect to caste determination, in bumblebees this occurs during larval development, meaning it is locked in prior to adult eclosion, which again makes the ability to mate that we have demonstrated extremely surprising from an evolutionary perspective. Finally, with respect to the last point, which addresses the ability of social insects to 'rear offspring', we agree with the reviewer that the ability to rear offspring is a hallmark of insect societies but emphasise the fact that worker mating and laying female-destined fertilised, diploid eggs, especially in corbiculate bees, is not

normal or expected in such superorganismal taxa. We believe that our revised manuscript makes this much clearer.

Reviewer Comment 2.2: Finally, the authors argue that their artificial-insemination result refutes an understanding that superorganismality cannot evolve without obligate worker un-matedness. This does not seem necessary. Superorganismality - i.e., eusociality - is a matter of degree, as measured by the level of worker reproductive altruism. Alas, un-matedness (and more generally, worker sterility) and eusociality co-evolve together, step by step. We don't expect superorganismality to suddenly appear upon the origin of obligate unmatedness, do we? If so, then I guess bumble bees were never superorganisms and the argument dissolves anyway. In total, despite the interesting success at coaxing workers to lay fertilized eggs and other interesting lab results, the author's narrative about resolving an evolutionary paradox or refuting theories about social evolution do to seem warranted.

Author Response 2.2:

As we understand the reviewer's argument, and apologies if this is not the case, it is based largely on treating 'superorganismality' as a synonym for 'eusociality'. In fact, both of these terms have a complex history and set of definitions. These have been comprehensively reviewed and put into a holistic theoretical framework by Boomsma (2022), and in this manuscript we are using this evolutionary framework, where 'eusociality' and 'superorganismality' are distinctly not the same thing. In fact, the transition to being a superorganism is a clear major transition in evolution, and thus explicitly a step-change rather than a matter of degree. We are sorry that we were not sufficiently clear in the manuscript about this. We believe that the additions we made in response to Reviewer 1 now make this distinction and theoretical framework clear.

Superorganisms (Boomsma 2022) have morphological-caste-differentiated colonies, where queens and workers are morphologically distinct as a result of a canalised developmental process, which should go hand-in-hand with lifetime worker unmatedness (Boomsma 2022). Consequently, given that the origin of superorganismality in bumblebees is dated to 25-40 MYA (Hines et al 2008), the retention of a functional spermatheca is both surprising and theoretically paradoxical. Our study and its results demonstrate that taxa can be both superorganismal (from the perspective of canalised development and morphological castes) and retain worker mating and sexual reproduction, which is not expected from comprehensive theoretical analysis, and so challenges current theory and calls for a deeper understanding of levels of organisation and supposedly irreversible transitions in social evolution (Boomsma 2022)(as described in the manuscript). Like reviewer 1, we believe that this means our work is of broad importance, both empirically and theoretically, in evolutionary biology.

Reviewers' Comments:

Reviewer #1:

Remarks to the Author:

I have now read this revised manuscript and am satisfied that all concerns I raised have been fully addressed in the revision. Overall, in my view this study has the strengths stated previously, representing a substantial, novel contribution to our understanding of the major transitions that should find a broad readership. My thanks to the authors for their revision and congratulations on their work.

I have a few, newly-arising minor comments, as follows:

1. Lines 38-42: The syntax needs attention here, since at the moment the text reads, 'Here, we show that ... (ii) the social conditions required for worker mating, ...'; i.e. point (ii) requires a verb.
2. Lines 266-269 (cf. Author response 1.3c): Thank you for suggesting adding the personal communication here, but as the point does not stem from my first-hand observation, I think the revised text as it stands is fine.
3. Line 390: The sense suggests that this subheading ('Sample ... extraction'), along with the following two, should be at a lower level than the one on line 383 ('Confirmation ... analysis').

Andrew Bourke

Reviewer #2:

Remarks to the Author:

I have read the revised manuscript and the response to reviewers. The discovery of mated workers in bumble bees is exceptional because apparently no one has noticed this before, despite *Bombus* being as relatively well-studied genus in social insect biology. The empirical work that follows this head-line result remains sound and informative, as I noted in my original review.

I noticed, however, that no changes were made to the manuscript in response to my comments, arguing instead that I am not up to date on current definitions or that I failed to agree with the other reviewer. I found this a bit disappointing and, in my view, is a missed opportunity to clarify for a general audience what broad evolutionary theory is actually being challenged or resolved.

Nonetheless, I see that in response to Andrew Bourke the authors did improve the conceptual narrative of their manuscript. If the authors are only making changes in response to Andrew, then please be sure to address his first major comment in earnest: "the ms states that the results are 'in contrast to previous theory', and it would benefit from being clearer as to exactly what theory or concept is being challenged...".

Would it be possible, for example, to state boldly that either Boomsma's criteria for superorganismality ('the theory') are incorrect, as evidenced by current findings, or that his criteria are correct and therefore Bumble bees and presumably many other taxa whose workers can mate, are not and never were superorganisms? In my view this bold but clear approach would be better than the wholly 'challenges current <unnamed> theory' vagueness.

Perhaps the authors could also up-date Boomsma's criteria rather than leaving the definitions issue hanging and unresolved.

Finally, if the focus is on how 'Bumble bees challenge current concepts in superorganismality' then the title of the ms could change.

Response to reviewer and editor comments (our responses indented and in italics)

REVIEWERS' COMMENTS

Reviewer #1 (Remarks to the Author):

I have now read this revised manuscript and am satisfied that all concerns I raised have been fully addressed in the revision. Overall, in my view this study has the strengths stated previously, representing a substantial, novel contribution to our understanding of the major transitions that should find a broad readership. My thanks to the authors for their revision and congratulations on their work.

Thank you for your time and for this positive feedback.

I have a few, newly-arising minor comments, as follows:

1. Lines 38-42: The syntax needs attention here, since at the moment the text reads, 'Here, we show that ... (ii) the social conditions required for worker mating, ...'; i.e. point (ii) requires a verb.

We incorrectly put the word 'that' before point (i) rather than after it, which is why it looked like point (ii) required a verb. We have now moved 'that' to its correct grammatical placement, and (ii) no longer looks like it requires a verb, see line39.

2. Lines 266-269 (cf. Author response 1.3c): Thank you for suggesting adding the personal communication here, but as the point does not stem from my first-hand observation, I think the revised text as it stands is fine.

Thank you - we, of course, respect your opinion here.

3. Line 390: The sense suggests that this subheading ('Sample ... extraction'), along with the following two, should be at a lower level than the one on line 383 ('Confirmation ... analysis').

Thank you for spotting this - we have now corrected it, see line 406, 417, 429.

Reviewer #2 (Remarks to the Author):

I have read the revised manuscript and the response to reviewers. The discovery of mated workers in bumble bees is exceptional because apparently no one has noticed this before, despite *Bombus* being as relatively well-studied genus in social insect biology. The empirical work that follows this head-line result remains sound and informative, as I noted in my original review.

Thank you for these positive comments

I noticed, however, that no changes were made to the manuscript in response to my comments, arguing instead that I am not up to date on current definitions or that I failed to agree with the other reviewer. I found this a bit disappointing and, in my view, is a missed opportunity to clarify for a general audience what broad evolutionary theory is actually being challenged or resolved.

Nonetheless, I see that in response to Andrew Bourke the authors did improve the conceptual narrative of their manuscript. If the authors are only making changes in response to Andrew, then please be sure to address his first major comment in earnest: "the ms states that the results are 'in contrast to previous theory', and it would benefit from being clearer as to exactly what theory or concept is being challenged...".

Would it be possible, for example, to state boldly that either Boomsma's criteria for superorganismality ('the theory') are incorrect, as evidenced by current findings, or that his criteria are correct and therefore Bumble bees and presumably many other taxa whose workers can mate, are not and never were superorganisms? In my view this bold but clear approach would be better than the wholly 'challenges current theory' vagueness.

Perhaps the authors could also up-date Boomsma's criteria rather than leaving the definitions issue hanging and unresolved.

Thank you for these very helpful comments. We have borne them in mind and, following additional instructions from the editor, made changes in the manuscript that we believe address all of these points. These changes and comments explaining them can be found in the revised manuscript as highlighted text associated with comments.

Finally, if the focus is on how 'Bumble bees challenge current concepts in superorganismality' then the title of the ms could change.

Thank you for this suggestion, which we have seriously reflected on. Given our explanation and discussion of our results and how they relate to organismality in the revised manuscript, we feel more comfortable keeping the title as it stands. However, we are happy to discuss this further with the editor if they feel we are not being strong enough.